# Vector-Borne Viral Diseases as a Current Threat for Human and Animal Health—One Health Perspective

**DOI:** 10.3390/jcm11113026

**Published:** 2022-05-27

**Authors:** Wojciech Socha, Malgorzata Kwasnik, Magdalena Larska, Jerzy Rola, Wojciech Rozek

**Affiliations:** Department of Virology, National Veterinary Research Institute, Al. Partyzantow 57, 24-100 Pulawy, Poland; wojciech.socha@piwet.pulawy.pl (W.S.); malgorzata.kwasnik@piwet.pulawy.pl (M.K.); m.larska@piwet.pulawy.pl (M.L.); jrola@piwet.pulawy.pl (J.R.)

**Keywords:** arboviruses, arthropod vectors, arbovirus transmission, vaccines, One Health perspective

## Abstract

Over the last decades, an increase in the emergence or re-emergence of arthropod-borne viruses has been observed in many regions. Viruses such as dengue, yellow fever, or zika are a threat for millions of people on different continents. On the other hand, some arboviruses are still described as endemic, however, they could become more important in the near future. Additionally, there is a group of arboviruses that, although important for animal breeding, are not a direct threat for human health. Those include, e.g., Schmallenberg, bluetongue, or African swine fever viruses. This review focuses on arboviruses and their major vectors: mosquitoes, ticks, biting midges, and sandflies. We discuss the current knowledge on arbovirus transmission, ecology, and methods of prevention. As arboviruses are a challenge to both human and animal health, successful prevention and control are therefore only possible through a One Health perspective.

## 1. Introduction

Climate changes, urbanization, growing trade, and international tourism associated with globalization have highly contributed to the expansion of blood-sucking arthropods into new territories [1]. As a result, the emergence or re-emergence of arthropod transmissible viruses (arthropod-borne, arboviruses) have increased significantly over the last decades [2]. Arboviruses are an ecological, polyphyletic group of viruses, mainly RNA, whose members belong to families such as *Flaviviridae*, *Peribunyaviridae*, *Phenuiviridae*, *Orthomyxoviridae*, *Reoviridae*, *Rhabdoviridae*, and *Togaviridae*. At least 300 types of mosquitoes, 116 species of ticks, and 25 species of midges are known vectors of arboviruses. In addition, sandflies, blackflies, stink-bugs, lice, mites, gadflies, and bedbugs have been shown to transmit arboviruses [3,4]. The CDC’s list of arboviruses includes more than 500 species, with more than 150 of them known to cause human and/or animal diseases [5,6]. Although arboviral infections may be asymptomatic or cause mild, transient influenza-like symptoms, they can be also associated with more severe consequences such as hemorrhagic fevers (e.g., dengue, yellow fever), encephalitis (e.g., Japanese encephalitis), or arthritis (e.g., Ross River fever, O’nyong-nyong fever, chikungunya) [7,8]. From a medical and veterinary point of view, diseases caused by arboviruses and their expanding geographical range are of major concern. Those are mainly mosquito-borne diseases such as dengue, zika, yellow fever, West Nile fever, Japanese encephalitis, chikungunya fever, or Rift Valley fever [9]. On the other hand, some arboviruses are still described as endemic, e.g., California encephalitis, Ngari, Nyando, or Pongola viruses; however, their range can increase in the foreseeable future [10]. A group of animal arboviruses can be distinguished, which, although important for animal breeding, are not a direct threat for human health, e.g., Schmallenberg virus (SBV), Akabane virus (AKAV), bluetongue virus (BTV), African swine fever virus (ASFV), epizootic hemorrhagic disease virus (EHDV), bovine ephemeral fever virus (BEFV), and African horse sickness virus (AHSV). Since arboviruses are a challenge to both human and animal health, therefore, successful prevention and control should be considered from the One Health perspective [11]. This review focuses on arboviruses infecting humans and animals and their major vectors: mosquitoes, ticks, biting midges, and sandflies. The current knowledge on arbovirus transmission, ecology, and methods of prevention are discussed.

## 2. The Main Arboviruses of Humans and Animals

The *flaviviruses* are currently regarded as the most important arboviruses from human health perspective, annually infecting as many as 400 million people. The most significant viruses from this family include: dengue virus (DENV), zika virus (ZIKV), yellow fever virus (YFV), West Nile virus (WNV), and Japanese encephalitis virus (JEV). *Flaviviruses* also infect a wide array of animal species and cause significant economic losses. Non-human primates are considered a reservoir for DENV, ZIKV, and YFV, whereas birds are the same for JEV or WNV. All these viruses can spill over to humans and sensitive animals [10,12]. The geographic distribution of *flaviviruses* includes Africa, South and North Americas, and Asia. West Nile virus has also spread to Europe and Australia, whereas Japanese encephalitis outbreaks are, so far, limited to countries in southeast Asia and the western Pacific. *Flaviviruses* could cause serious clinical symptoms such as hepatitis, vascular shock syndrome, encephalitis, acute flaccid paralysis, congenital abnormalities, and abortion [13]. *Togaviridae* is another important family of arboviruses infecting humans, with the Chikungunya virus (CHIKV) being one of the most prominent. Natural hosts of CHIKV are wild primates. Cases and outbreaks of the diseases caused by infections with CHIKV have been recorded in Africa, Asia, Europe, the Indian and Pacific Oceans regions, and, since 2013, in the Americas. In 2021, more than 220,000 cases of CHIKV infections were reported in the world, with Brazil and India being the most affected countries [14]. Rift Valley Fever virus, a member of the *Phenuiviridae* family, is also considered a high epidemic threat for humans. It mainly circulates in Africa, although outbreaks in Saudi Arabia and Yemen have been reported. The RVFV infects primarily domestic animals (sheep, goats, and cattle), causing high infant mortality and abortion [15]. Historically, in humans, infections with RVFV led to large outbreaks, with one of the largest in 1977 in Egypt, causing over 200,000 cases and 600 deaths [16]. In the last two decades, over 4000 human cases have been described, resulting in around 1000 fatalities. The current number of cases could be even higher, as surveillance for this virus in humans is lacking [17]. Human infection results in a wide range of clinical symptoms, from self-limiting febrile illness to life-threatening hemorrhagic disorders and abortion. Seropositive cases detected in Turkey and Tunisia and outbreaks in a Mayotte (French overseas department) raised concerns in the EU about possible incursions into countries neighboring the continental EU or having direct trade contacts [18,19].

There are a group of arboviruses affecting animals without known threats to humans. Those viruses, which are of economic importance and may affect animal welfare and the international trade of livestock, include members of the *Reoviridae* family such as BTV, AHSV, and EHDV. Bluetongue virus affects wild and domestic ruminants such as sheep, goats, cattle, buffaloes, deer, most species of African antelope, and camels. During the last two decades, there has been a global expansion of BTV distribution, and novel viral serotypes have been identified. Currently, at least 27 serotypes of BTV are known and the presence of reassortants of viruses circulating in Europe and the United States has been confirmed [20]. African horse sickness virus infects equids including horses, mules, donkeys, and zebras. Infections of AHSV continue to occur regularly in southern African countries, but the virus has also been occasionally detected in North Africa, the Middle East, the Arabian Peninsula, and Asia [21]. The most recent outbreak outside of Africa occurred in Thailand in 2020, a country previously recognized as AHSV free [22]. Epizootic hemorrhagic disease virus primarily infects wild ruminants (in particular white-tailed deer) but has been isolated also from domestic ruminants. Infections with EHDV have been reported in North America, South America, Asia, Africa, Australia, and, more recently, in countries surrounding the Mediterranean Basin (Morocco, Algeria, Tunisia, Israel, Jordan, and Turkey). Epizootic hemorrhagic disease virus demonstrates immunological cross-reactivity with BTV [20,23]. Schmallenberg virus and AKAV are members of the family *Perib**unyavirida**e*, the Simbu serogroup. These viruses cause congenital abnormalities of the central nervous system (CNS) and musculoskeletal system in ruminants [24]. The Schmallenberg virus emerged at the border region between Germany, the Netherlands, and Belgium in 2011 and was the first identified European member of this serogroup [25,26]. Akabane Virus is present in Asia, Africa, the Middle East, and Australia. Countries that have reported AKAV are Japan, Korea, Taiwan, Australia, Israel, and Turkey [24]. African swine fever virus (ASF) is a DNA arbovirus belonging to the family *Asfarviridae*. It affects domestic pigs and wild boars, causing fatal hemorrhagic disease. It was originally identified in Kenya in 1921 and then confirmed in 35 African countries. After the first confirmation of ASFV in the 1950s in Portugal, it is now re-emerging in Europe. In August 2018, ASFV was reported in Asia, where it has continued to spread [27]. In Europe, between February 2021 and January 2022, 1912 outbreaks of ASF were reported in domestic animals (majority in Romania—1590) and 8490 cases in wild boars (majority in Poland—4669) [28]. Lumpy skin disease virus (LSDV) and bovine ephemeral fever virus (BEFV) representing *Poxviridae* and *Rhabdoviridae*, respectively, affect cattle and wild ruminants. They are highly infectious viruses but cause low mortality. Lumpy skin disease was initially limited to Africa but spread to the Middle East in the 1990s and has recently spread to Europe and Asia [29,30]. Bovine ephemeral fever virus occurs throughout tropical and subtropical regions of Africa, Asia, and Australia [31,32]. Additional information on the selected arboviruses is summarized in Table 1.

## 3. Transmission of Arboviruses

Transmission of arboviruses can occur either vertically or horizontally. The ability to combine those different modes of transmission enables arboviruses to survive under adverse conditions [61,62].

### 3.1. Horizontal Transmission

Horizontal transmission (HT) encompasses all modes of non-parental transmission, including biological and non-biological transmission [63]. The biological transmission mechanism is unique for arboviruses and is required to sustain the replication cycle [64]. It requires a high density of competent vectors, a high vector survival rate, and frequent contact between the vectors and vertebrate hosts [65]. The match between the virus, the competent vector, and the susceptible vertebrate is important for effective biological transmission [61]. Compatibility between a virus and vector relies on the presence of specific receptors on arthropod cells. After entering the midgut with a blood meal consumed by the vector, a virus must cross the gut barrier and then replicate in the insect body before being released [61]. For this process to be completed, arboviruses have to cope not only with the physical barrier of midgut epithelial cells but also with immunological and biochemical barriers that include proteolytic enzyme upregulation, RNA interference, peritrophic matrix formation, and antimicrobial molecule influx [66]. The combined effect of these mechanisms, and the requirement for efficient transmission to vertebrates, result in recurring bottlenecks that arboviruses undergo in their replication cycle, affecting its evolution [67]. Viremia detected in vertebrates is not an obligatory condition for the biological transmission of arboviruses. Studies on *Thogotovirus* infections showed that a vertebrate host, free of viremia, may play a role in arbovirus epidemiology through a process called “nonviremic transmission” (NVT). This mechanism was described in vivo when naïve guinea pigs were used for simultaneous feeding of virus-infected and non-infected ticks (adult and nymphs). As a result, a high percentage of the previously uninfected ticks became infected, while no viremia was detected in host animals [68]. Although knowledge of the importance of NVT in natural conditions is still limited, it appears that reduced pathological impact on the vertebrate host could make transmission more effective [69,70].

It is postulated that the detection of the virus in the salivary glands is biological evidence of the infection and indicates the competence of the vector. The detection of the virus in the homogenized body of the insect may only indicate the ingestion of blood meal from a virus-infected host [71]. Arthropods’ saliva serves not only as a passive carrier of the virus but also promotes virus transmission via specific salivary molecules in a process called Saliva Assisted Transmission (SAT). This phenomenon has been observed for various blood-sucking arthropods [72,73]. In the case of fleas, mosquitoes, and midges, the duration of blood-feeding is a matter of minutes, for soft ticks, hours, and up to two weeks for hard ticks [74]. Prolonged feeding of ticks means that they must cope with the host’s inflammatory and immune responses. Their saliva has been confirmed to play an important role in immunomodulation and mediating of Tick-Borne Viruses’ (TBV) transmission [75,76]. Tick saliva exhibits cytolytic, vasodilator, anticoagulant, anti-inflammatory, and immunosuppressive activity, whereas other salivary gland secretions include cementing substances that help attach the parasite to the host skin and hygroscopic components that play a role in absorbing moisture from the environment [73]. Fast-feeding mosquitoes and sandflies also produce salivary factors modulating the host’s immune defense, but their effect is observed after the arthropod leaves the host [74]. A vertebrate host can feed a large number of biting arthropods, therefore, an immune response to arthropod salivary factors, induced by earlier exposure, may be detrimental to the later feeding of the arthropods [77,78]. Mosquito saliva also has an immunomodulatory effect that inhibits the host’s ability to respond to infection. Studies on the arboviruses in in vitro systems showed that their replication may be promoted by the addition of arthropod salivary components, which may suppress the production of antiviral cytokines [79,80].

Non-biological transmission can be direct (non-vector mediated) or mechanical (through a vector mouthpart, without replication in an arthropod). Direct transmission may be intranasal, oral (or nasopharyngeal), or venereal or by exposure of cornea, mucous, or skin with an abrasion. The possibility of venereal transmission of arboviruses, also between vectors, has been described, e.g., zika virus transmission in *Aedes aegypti* mosquitoes [81], Chandipura virus in *Phlebotomus papatasi* (Scopoli) [82], Vesicular Stomatitis Virus in *Culicoides sonorensis* [83], and ASFV in *Ornithodoros (Pavlovskyella) marocanus* soft ticks [84]. The common sources of pathogens include food, aerosol, bodily secretions, urine, feces, saliva, milk, hair, feathers, and skin [61]. Worthy of note is the fact that insectivorous animals, such as bats, could become infected with arboviruses by the consumption of infected mosquitoes. This type of transmission has been confirmed for RVFV, YFV, and JEV [85]. Mechanical transmission takes place when an arthropod vector with contaminated mouthparts bites the vertebrate host. This does not require amplification of the virus within the vector [86]. Ticks and mites, compared to other insect vectors, are not efficient mechanical transmitters due to their low frequency of intermittent feeding and remaining on the same host for each life stage [61].

### 3.2. Vertical Transmission

Vertical transmission (VT) refers to the transmission of a pathogen from parent to offspring (transgenerational transmission) or sustained infection between different development stages of arthropods (transstadial transmission). Transgenerational transmission usually occurs transovarially (through the ovaries) after the virus infects the ovarian germinal tissue and then is transmitted transstadially to the next reproductive or blood-feeding stage. Arboviruses could also be transferred to the next generation through transovum transmission, in which the virus remains on the surface of the egg after being laid by an infected vector [87]. Transgenerational transmission has been observed in ticks, mosquitoes, midges, and sandflies [63,88,89,90]. To make virus persistence possible, transgenerational transmission in vectors must also include transstadial transmission to ensure maintenance of the infection through the reproductive, blood-feeding adult stage [87]. Transstadial transmission is especially important in the case of the ticks, as each development stage requires separate blood meal at which it can become a vector that could pass viruses through the next stages of development [90].

Although VT frequency is estimated at below 1% for many viruses, including *flaviviruses*, for some, such as *orthobunyaviruses*, it can be much higher (16–28%) [63]. It is believed that VT in arthropods is a means of pathogen survival under conditions that are unfavorable for horizontal transmission. However, statistical studies have shown that VT cannot be the main mode of transmission, no matter how efficient it is, and therefore requires coexisting horizontal transmission [61].

## 4. Main Arthropod Vectors of Arboviruses

Arthropod vectors include mosquitoes, ticks, biting midges, flies, mites, fleas, bedbugs, lice, and other arthropods that are capable of transmitting pathogens from one host to another. They are characterized by their ability to tolerate high levels of viral proliferation without developing significant pathology [91]. Nevertheless, it was documented that the introduction of the arbovirus to the vector body leads to transcriptomic and proteomic changes. Additionally, as it was proven in the example of DENV infecting the *Aedes aegypti* mosquito, arboviruses could also affect vector behavior by increasing the frequency of biting, which results in a higher probability of virus transmission [92]. Analysis of vectors collected from different geographic regions confirms that their genetic variation can often lead to differences in the activity of proteins that can determine their susceptibility and competence [93,94]. Vectorial capacity, defined as the efficiency of arbovirus transmission, depends on vector competence, the density of vectors and hosts, and environmental conditions [95]. Typically, arboviruses are associated with specific vectors, however, some, such as WNV or JEV, can be vectored by many different mosquito species, ticks, and other arthropods [96]. In this section, we focus on those vectors considered as the most significant for arboviral transmission: mosquitos, ticks, biting midges, and sandflies.

### 4.1. Mosquitoes

Mosquitoes are considered the most important arthropod vectors of arboviruses in almost every part of the world. Different species of mosquitoes are present on every continent except Antarctica. Although mosquitoes are distributed throughout the world, generally, in areas with a higher temperature and humidity, larger populations and greater diversity are observed. Mosquito-borne diseases are an increasing problem, not only in tropical regions, where fast urbanization, migration, and population growth are occurring, but also in new and, so far, unaffected areas of North America or Europe [95].

Mosquitoes are insects belonging to the order Diptera, suborder Nematocera. Species of particular medical importance belong to the family *Culicidae*, which includes about 110 genera and subgenera and about 3600 species. The majority of mosquito species responsible for the transmission of arboviruses belong to the genera of *Culex*, *Aedes*, and *Anopheles* [97,98,99,100]. *Aedes* mosquitoes, especially *Aedes aegypti* and *Aedes albopictus* (Asian tiger mosquito) transmit dangerous viruses from families such as *Flaviviridae*, *Togaviridae*, and *Phenuiviridae* including CHIKV, DENV, YFV, and RVFV [101]. It is reported that *Aedes* spp. distribution is now the widest ever recorded, with more than three billion people noted as currently living in *Aedes*-infested regions. *Aedes albopictus* is now described as the most invasive mosquito in the world [102,103].

Mosquitoes have four development stages: egg, larvae, pupa, and adult (imago). The first three stages are related to the aquatic environment, whereas the adults are flying insects (Figure 1a). A mosquito’s life cycle takes approximately 8–10 days. The average life expectancy of a female mosquito is about 6 weeks, but under favorable conditions, it can last up to 5 months. Many mosquito species naturally undergo multiple reproductive cycles (blood-feeding and egg-laying) in their lifespan [104]. The female mouthparts form a long piercing/sucking proboscis, whereas males are not adapted to piercing the skin. Male mosquitoes feed on nectar from flowers; only females feed on the blood of humans or other animals when they are ready to produce eggs. All mosquito species feed using the same physical mechanisms, although different feeding timetables and behaviors are observed. Two thin serrated cutting projections called maxillae are used to pierce the outer epidermis into the layer of the dermis below. Between the maxillae, there is a structure of two connected tubes, one for blood collection and the other for the introduction of saliva into the host. The labrum has sensors to detect blood vessels and, finally, blood is drawn from the vessel. On top of the labrum, there is the lower part of the hypopharynx through which saliva is continuously secreted [41].

### 4.2. Ticks

Ticks are obligatory, blood-feeding ectoparasites of human and other vertebrates with worldwide distribution. The range of geographical distribution of ticks is affected by occurring climate changes, as it is for other vectors of arboviruses. However, in the case of ticks, this effect is slow, indirect, and independent from any short time climate variability. This is due to a tick’s ability to find shelter from extreme weather conditions in soil and their limited mobility compared to mosquitos and other short-living flying vectors. For example, observed poleward expansions of ticks in Sweden seem to be connected with the higher survival rate of arthropods but would be impossible without an accompanying increase in range and population of important tick host species, such as roe deer [105,106]. Ticks are Arachnids belonging to subclass Acari, order Parasitiformes, and suborder Ixodida. There are at least 899 recognized species of ticks belonging to three families: *Argasidae* (soft ticks—containing 195 spp.), *Ixodidae* (hard ticks—containing 703 spp.), and *Nuttalliellidae* (1 sp.) [107]. It is estimated that less than 10% of those are known to be viral vectors. Currently, around 50 viral species representing one DNA virus family (*Asfarviridae*) and eight RNA virus families (*Flaviviridae*, *Orthomyxoviridae*, *Reoviridae*, *Rhabdoviridae*, *Nairoviridae*, *Nyamiviridae*, *Phenuiviridae*, and *Peribunyaviridae*) have been identified as transmitted by ticks. Almost 25% of TBV are described as pathogenic for humans and animals [75]. Several TBV, such as Tick-borne encephalitis virus (TBEV) or Crimean–Congo hemorrhagic fever virus (CCHFV), are regarded as a significant threat to human health as they are common and could cause serious diseases. Two main tick families have different lifecycles and ecology affecting their ability to function as a vector. Soft ticks transmit fewer human pathogens, probably due to short feeding times (from minutes to hours) and their restricted habitats (such as caves, wildlife nests, or burrows). In contrast, hard ticks are found in different habitats, including urban and suburban areas; they could feed on one host for many days, depending on species and tick development stage. As a result, they more often become the source of infection in humans [108]. Ticks are characterized by a complex lifecycle including, in most cases, four stages: egg, larvae, nymph, and adult form (Figure 1b,c). For both males and females, progress between each phase of development requires separate blood meals [97]. In most tick species, each of the three active forms (larvae, nymph, and adult) feeds on separate hosts but, in some cases, two or only one host(s) are required. If the lifecycle requires multiple hosts, they often represent different vertebrate groups. For example, for immature forms (larvae and nymph), it is common to feed on small animals such as birds or rodents, whereas adults more often feed on large carnivores or ungulates. Although some ticks have a high preference for particular host species, currently, no ticks that strictly require humans to complete their lifecycle are known. As a result, tick-borne diseases are exclusively zoonoses, with most tick-borne pathogens relying on wild animals as their reservoir, whereas humans only become incidental hosts. Depending on the climate zone, ticks could pass through all development stages in one year or, in the colder regions, it can take 3–4 years. This longevity, and the fact that usually, once infected, they become lifelong carriers, make them important reservoirs of arboviruses [75,90].

In both soft and hard ticks, the mouthpart, called chelicerae, can be distinguished. Its design enables cutting through the epidermis and dermis of the host by a sawing motion which leads to the creation of a blood pool [97]. Ticks are long-term pool feeders, meaning that they may feed on their host for hours or even many days by absorbing fluids from the hemorrhagic pool [73]. Transmission of viruses from infected ticks could occur in a relatively short time, spanning from 15 min after attachment to the host, as in the case of the Powassan virus (family *Flaviviridae*), 3 h for TBEV, or within 24 h in the case of the Thogoto virus (THOV, family *Orthomyxoviridae*) [75,109].

### 4.3. Biting Midges (Culicoides)

Biting midges are widely spread through different geographical areas, from America to Europe, Asia, and Oceania, with the exception of Antarctica and New Zealand [110,111]. They are among the smallest blood-sucking flies, with body lengths that rarely exceed 3 mm [111]. The range of *Culicoides*-borne diseases are closely related to climate. In regions of temperate climate, the seasonality of virus transmission coincides with warm, humid summer and fall periods. In tropical and subtropical regions, high infection rates are recorded during wet summers [112]. It is reported that more than 50 arboviruses, belonging to, among others, the *Peribunyaviridae*, *Reoviridae*, or *Rhabdoviridae* families, were isolated from different *Culicoides* species [111].

The genus *Culicoides* (order Diptera, family *Ceratopogonidae*) currently contains 1347 species in 33 subgenera and 38 species groups [113]. The subgenus *Avaritia* contains more vector species than other groups [114,115]. *Culicoides imicola* is widespread in the world and can transmit a number of different arboviruses important for animal health, such as BTV, EHDV, or AHSV [116,117]. The ability of *Culicoides* to transmit SBV was confirmed in laboratory conditions [118]. The role of species other than *Culicoides imicola* was proven for the first time in Europe by a study from Italy [119]. The predominant *Obsoletus* complex and *Pulicaris* complex were implicated in BTV transmission during the outbreak of bluetongue in northern Europe in 2006 [120], whereas *Culicoides sonorensis* was confirmed as a vector of EHDV in North America [121]. The role of *Culicoides* in ASFV transmission is also considered [122]. Although Akabane virus (AKAV) was initially isolated from mosquitoes, it has been reported that the main vectors for AKAV are the various *Culicoides* species [91,112]. Some *Culicoides*-borne viruses, such as Oropouche virus (OROV) and its reassortants, including Iquitos virus and Madre de Dios virus, are also human pathogens [123,124,125]. Those viruses circulate in South and Central America [125].

The life cycle of biting midges includes an egg, four larval stages, a pupa, and an adult form (imago) (Figure 1d). The development of *Culicoides* can last from several weeks to several months. The larvae are able to survive in unfavorable environmental conditions (they overwinter), so the length of the process depends on the ambient temperature. Almost all *Culicoides* require moisture-rich habitats for their development, and this determines their population and seasonality [111]. Adult individuals are short-lived; only a few can survive longer than 10 to 20 days. During this period, females are capable of feeding on the host multiple times [111]. Similarly, as in mosquitos, *Culicoides* females depend on blood for the maturation of the eggs, although, in a few species, they are autogenous and therefore may produce an initial batch of eggs without feeding, using only reserves stored from the larval period [126]. Males do not feed on blood; they can survive on nectar alone. Female mouthparts make a proboscis, perfectly suited for cutting skin and sucking blood [126]. Wings are well developed, and biting midges are commonly identified at the complex or species level based on the wing maculation [127]. Their hosts, depending on the species, are mammals and/or birds [111].

### 4.4. Sandflies

Various sandfly species have been distributed throughout the world in tropical, subtropical, arid, and semi-arid areas as well as temperate zones of southern Europe, Asia, Africa, Australia, and Central and South America. As the geographical distribution of sandflies is restricted by climate conditions, such as sufficiently high moisture or average temperature of the coldest month (not lower than 5 °C), climate changes observed in recent years may promote the spread of this vector and, in consequence, increase the risk of infection with certain arboviruses. In Europe, this risk is considered especially high on the Atlantic Coast and in parts of Germany, Switzerland, Hungary, and Austria [128]. Sandflies are blood-feeding arthropods belonging to suborder Nematocera within the order Diptera and are members of the *Phlebotominae* subfamily within the *Psychodidae* family, consisting of nearly 1000 member species. Among six described genera, two *Phlebotomus* present in the “Old World” and *Lutzomyia* sensu lato, confined to the Americas, are regarded as important for their ability to transmit causative agents of human and animal diseases [129]. Sandflies are known as vectors of viruses belonging to three different genera: *Phlebovirus*, (family *Phenuiviridae*) including Sandfly Fever Sicilian virus, Sandfly Fever Naples virus, Toscana virus, and Punta Toro virus; *Vesiculovirus* (family *Rhabdoviridae*), including Chandipura virus; and the *Orbivirus* (family *Reoviridae*), including Changuinola virus [130,131]. Phlebotomine sandflies are small with body lengths rarely larger than 3 mm. Whereas both males and females feed on sugar sources such as sap of plants and honeydew of aphids, females also feed on blood which is, in most cases, required for the production of eggs. Depending on the species, the number of bloodmeals needed for each gonotrophic cycle may vary, from multiple to only a single for each batch of eggs. Their lifecycle encompasses an egg and four larval stages followed by pupa and the adult stage (image) (Figure 1e). Most sandflies are active in the evening and at night, with only a few species biting during the daylight. When non-active, they rest in cool and humid places such as houses, latrines, cellars, stables, caves, fissures in walls, rocks or soil, dense vegetation, tree holes and buttresses, burrows of mammals, bird’s nests, and termitaria [132]. Although the mouthpart of sandflies is similar in structure and function to a mosquito’s proboscis, its feeding mechanism is more similar to that observed in ticks, with a blood pool formed under the dermis of the host. However contrary to ticks, extraction of blood is rapid, with feeding time lasting minutes [97].

## 5. Ecology of Arboviral Infections

Transmission of the arboviruses occurs under certain environmental conditions and is related to three components: the vertebrate host, the arthropod vector, and the virus. As described by Weaver et al. in 2021, arbovirus transmission involves ecologically distinct cycles: the sylvatic (enzootic), the urban (human-amplified), or the rural (epizootic) (Figure 2) [67]. Arboviruses may circulate in the sylvatic cycle with rodents, birds, or non-human primates acting as reservoir hosts. These wild animals usually do not show clinical signs of infection but could become viremic and contribute to the maintenance of the virus in the ecosystem [133]. For many arboviruses, humans and domestic animals are considered dead-end hosts. This means that although they are susceptible to infection with these viruses, they usually develop only limited viremia, with viral loads in blood not high enough for effective transmission to vectors. As a result, the viral transmission cycle is stopped, even though clinical symptoms can occur [134]. However, some arboviruses cause high levels of viral load in humans and can be transmitted between humans by mosquitoes. Viruses found in African primates (e.g., YFV, ZIKV, and CHIKV) or Asian primates (e.g., DENV) were all initially transmitted by forest canopy mosquitoes. However, the ability of anthropophilic mosquitoes such as *Aedes aegypti* to transmit these viruses has resulted in humans becoming a new host and initiation of the urban cycle [135,136]. When a virus from the sylvatic cycle enters urban environments, an infection can spread rapidly through highly anthropophilic urban mosquitoes [137,138]. Environmental changes undoubtedly increased the importance of urban cycles of arboviruses. Humans are at risk of infection when they enter forest habitat, through deforestation, hunting, agriculture, or urbanization. On the other hand, imported zoonoses could potentially “spillback” into local wildlife [139,140]. Such a threat is posed by urbanization which favors contact between human and wild animal populations. The risk associated with the spreading of arbovirus in urban cycles is also increased by improper waste management and accumulation of trash (used automobile tires, plastics, tins, etc.), which creates favorable ecological conditions for species such as *Aedes* populations to reproduce and thrive [136]. A rural cycle occurs when domesticated animals are used as (secondary) amplification hosts and is associated with increased viral spillover to humans in agricultural settings. This kind of cycle is typical for the viruses such as Venezuelan equine encephalitis virus (VEEV) amplifying in equines, JEV in swine, or RVFV in ruminants [141]. The urban cycle is more important for human medicine as large urban centers are more prone to massive spread of infection. Rural cycles in areas with high livestock densities are the focus of veterinarians’ attention. However, since there are viruses that can follow both cycles, they should be viewed together from the One Health perspective.

## 6. Preventive Measures for Controlling Vector-Borne Viral Diseases

Since the first evidence of the role of arthropods as vectors of viral diseases, various preventive approaches have been introduced, starting with the campaign against yellow fever in Central and South America at the beginning of the 20th century [142]. Early prevention programs targeted mosquitoes and, so far, most of the actions have been directed against these vectors. This is due to their intense geographic expansion and the importance of the viruses they transmit and, thus, the high risk to human life [95,142]. Practices currently used for the prevention and control of arboviral disease spread could be divided into two main categories: those targeting arthropod vectors and those directed to viruses themselves (through vaccines or therapeutics). The first approach aims to decrease the local population of arthropod vectors, limit the probability of contact with potential hosts, or suppress their ability to transmit viruses. This could be achieved by using one of the established types of control methods: environmental, mechanical, biological, and chemical, supplemented by recently developed genetic methods (Figure 3) [143,144]. Those methods, in most cases, are not mutually exclusive and could act as complements of each other. The choice of particular methods or their combination should be based on Integrated Vector Management (IVM)—defined as a rational decision-making process to optimize the use of resources for vector control. This involves evaluation of cost effectiveness and ecological sustainability of prevention through collaboration with the health sector and local communities [144].

### 6.1. Environmental Prevention

Environmental methods are directed to change local habitats in such a way that they become less favorable for the breeding or survival of a particular vector. This proved to be especially effective for limiting the mosquito population in urban areas through so-called “source reduction”. This includes simple interventions limiting accessibility to the potential aquatic breeding areas such as catch basins, drums, open water tanks, old tires, and other containers [145,146]. Historically, source reduction directed against *Aedes aegypti* and *Aedes albopictus* was one of the crucial parts of the successful preventive strategies against epidemics of yellow fever in South and Central America in the first half of the 20th century and dengue in Cuba in the 1980s [142,147]. As the lifecycle of ticks is long, often lasting years, and they are able to survive in harsh weather conditions, environmental interventions that could be effective in reducing their population are difficult [148]. The specificity of the biology of ticks requires that each intervention be applied repeatedly and over a longer period of time. Additionally, unlike mosquitoes, tick breeding sites are more difficult to locate and remove [108,148]. However, as ticks require blood-feeding at each stage of their development, targeting potential wildlife hosts could also reduce the population of a vector. For example, in the USA, reduction in the local population of *Ixodes scapularis* tick was achieved through lowering the density of one of the main tick hosts—white-tailed deer [149,150]. Similar to ticks, source reduction in the sandfly population is difficult as the knowledge of breeding places and habitats of their immature stages is limited for most of the known species [151,152]. Some local achievements in limiting the adult population of one of the sandflies, *Phlebotomus argentipes*, was achieved through simple measures such as plastering of holes in homes and cattle sheds where breeding sites were identified [153]. In general, for the environmental interventions to be effective, cooperation with local communities is crucial as, for example, mosquitoes’ source reduction requires removal of all potential breeding sites including those located on private proprieties. This is not possible without appropriate educational programs and/or hiring qualified professionals for surveillance [143].

### 6.2. Mechanical Prevention

Mechanical methods of vector control are based either on the use of various traps designed to lure and kill vectors or the application of bednets and meshes (insecticide infused or not) to limit the chance of successful feeding and pathogen transmission [154]. Infusion with insecticides seems to be especially important for prevention against *Culiccoides* and sandflies as those vectors are small enough to pass through meshes and nets effective against mosquitoes, so a purely mechanical barrier could be ineffective [155,156]. Although traps are usually used only for vector monitoring, field tests performed in Peru and Brazil using specially designed lethal ovitraps (traps luring and killing egg-bearing mosquito females) showed that they could also be effective means of reducing the *Aedes aegypti* vector population and potentially lead to lowering the number of dengue fever cases [157,158]. A similar approach was tested for limiting the population of *Rhipicephalus sanguineus* ticks, regarded mainly as a dog parasite but also able to transmit viruses (e.g., THOV) to humans. In that case, sticky traps baited with slowly released pheromones were found to reduce tick infestation in dog kennels [59,159].

### 6.3. Chemical Prevention

Chemicals have been used widely for vector control since the development of DDT in the 1940s. Large-scale use of this insecticide in the 1950s and 1960s, accompanied by source reduction, was one of the bases of success of the South American *Aedes aegypti* mosquito eradication program that led to effective control of yellow fever and dengue in the area [160]. Currently, two main categories of chemical insecticides are used in vector control: larvicides (targeting the immature stages) including insect growth regulators (IGRs, e.g., pyriproxyfen, methoprene, diflubenzuron) and adulticides. IGRs poses ovicidal and larvicidal proprieties and are characterized by relatively low toxicity to humans, as they affect insect-specific development processes. To be effective, IGRs are used to treat breeding sites of arthropod vectors. Adulticides are directed against adult forms and are distributed either through ground spraying, indoor residual spraying (IRS) of walls of human dwelling, the use of insecticide-treated materials (ITMs) (such as bednets), or attractive toxic sugar baits (ATSBs). The choice of individual strategies depends on the desired goal, as they can be used for prevention but also for outbreak control. For example, ground spraying with pyrethroids is effective in rapidly reducing the number of female mosquitoes and thus effectively reducing the scope of the epidemic. However, in some countries, its prophylactic use is limited due to the toxic effects [143]. It was found that insecticides directed against mosquitos are also effective against other insect vectors such as sandflies and biting midges [156,161]. However, the choice of chemicals and their concentration and application techniques should be adapted to targeted vector species. For example, standard approach for control of mosquito population is distribution of adulticides as ultra-low volume sprays during the nighttime. While this approach is effective against *Anopheles* species (most active from dawn to dusk), it is less efficient for mosquito species with diurnal activity habits (e.g., *Aedes aegypti* or *Aedes albopictus)*. This could potentially be overcome by the use of the adulticides that contain excitatory substances that force *Aedes* mosquitos to leave their nighttime resting places, making them vulnerable to lethal aerosol [162]. On the other hand, this problem does not concern sandflies, as they exhibit nocturnal activity and standard spraying procedure with pyrethroids could be adapted. However, as sandflies are terrestrial breeders and their larvae are difficult to locate, control measures are usually limited to those targeting adult individuals [156]. The use of chemical prevention against ticks, apart from aerosol spraying, includes baited traps designed to lure potential small mammalian hosts. Inside the traps, animals are brushed with acaricids. This method was found to effectively reduce the population of larval stages of ticks [163]. Although chemical control methods proved to be effective in vector-borne disease control, there are growing concerns about their further use due to their potential negative effect on human health and the emergence of resistance in targeted vector populations [164].

### 6.4. Biological Prevention

Biological control methods use natural predators, parasites, or pathogens of vectors to reduce their population or affect their ability to spread arbovirus infections [165]. They are considered as a potential alternative for pesticides as they are regarded as safer for the environment, cheaper, and less prone to become ineffective by the development of resistances in the population of targeted vectors [166]. A number of entomopathogenic fungi are available commercially, including two, *Beauveria bassiana* and *Metarhizium anisopliae*, that were proved to be effective in the control of the population of ticks, mosquitos, and biting midges. Although there were found to be safe for the environment and non-target species, their efficiency was highly dependent on weather conditions (reduced in high temperature and low humidity), selected fungi strains, formulation procedure, as well as application and delivery methods [143,167,168]. Additionally, they have to be carefully chosen for particular use, e.g., *Lagenidium giganteum* efficiently infects larvae of *Aedes aegypti* and *Culex pipiens* leads to high mortality but they have no effect on *Anopheles gambiae* [166]. The efficiency of treatment also varies between groups of vectors. For example, while entomatopathogenic fungi were found to cause high mortality against each development stage of ticks, they were found to be effective predominately against larval stages of mosquitoes [167,169].

Natural predators of vector species could serve as another mode of biocontrol, as long as they are specific only to selected vectors, do not compete with local species, can be easily grown in artificial conditions, and are introduced in large quantities into the environment [170]. The oldest example of biological control of mosquitoes dates back to early 1900s when larvivorous fishes such as *Gambusia affinis* or *Poecilia reticulata* were artificially introduced as a part of control programs [171]. More recently, the use of cyclopoid copepods (*Megacyclops viridis*), the most effective invertebrate predator of mosquito larvae, was proposed. It was shown both in laboratory settings and field trials performed in Asia and North America that introduction of those invertebrates into water containers inhabited by mosquito larvae could significantly reduce the number of adult vectors or even lead to local eradication of mosquitos [143]. On the other hand, although natural predators of ticks, such as certain species of birds including oxpecker (*Buphagus* spp.) or parasitoid wasp and flies, are known, they are impractical to use for biological control as they are either not effective enough in reducing tick population or it is difficult to increase their population artificially [170].

Another category of biological prevention methods is based on bacterial toxins that could act as natural larvicides (e.g., products of *Bacillus thuringiensis svar. israelensis (Bti)* and *Lysinibacillus*
*sphaericus*) as well as a fermentation product of *Saccharopolyspora spinosa* (Spinosan). They can be used either separately or in combination, which decreases the risk of resistance of insects and enables a wider target range [172].

A promising alternative method of biocontrol is the application of the endosymbiotic bacterium *Wolbachia pipientis*, found in over 70% of known insect species [173]. It infects the gonads of the host and could be transmitted to the next generation via female adults to their eggs [143]. It was found that this *Wolbachia* could be useful for controlling mosquito-borne viral diseases spread via a few different mechanisms. First, infection with *Wolbachia* could lead to reproductive alteration, called Cytoplasmic Incompatibility (CI). CI means that reproduction between uninfected females and infected males or mosquitos infected with different strains of the bacterium is not possible due to embryogenic mortality. This could be used for population suppression by the repeated introduction of *Wolbachia* infected males [143]. Efficiency of population suppression with *Wolbachia* is highly dependent on proper sex sorting, which increases the cost of this approach [174]. It was also found that certain strains of *Wolbachia* could cause a reduction in the lifespan of infected insects. Mosquito females with reduced lifespan are less likely to serve as a vector for viral diseases, as any pathogen requires a certain period of time needed for replication in insect cells and reaching the salivary gland (so-called extrinsic incubation period). For example, only mosquitos older than 10–14 days could be effective vectors of DENV [173]. Finally, it was found that some strains of *Wolbachia* could interfere with various human pathogens, including DENV or CHIKV [168]. Currently, the most promising results have been obtained in the use of *Wolbachia* in limiting the spread of DENV by *Aedes aegypti*. In Indonesia, the sharp reduction in the rate of Dengue fever cases was observed following the release of *Wolbachia* infected mosquitoes [175]. It was found to be a highly cost effective method which could give even better results when combined with vaccines [176]. Although viral evolution could lead to selection of resistant strains, multiple mutations in the viral genome would be required to overcome the broad mode of action of *Wolbachia*, without losing virus competence to both human and insect tissue [177]. As it was found that *Wolbachia* also infects sandflies and midges, control methods described above could also be adapted for controlling viruses spread by those vectors [178,179].

### 6.5. Genetic Prevention

Genetic control methods are based on the mass release of genetically modified insects into the environment. Modifications could lead to population suppression achieved through introduction of sterile males (so-called Sterile Insect Technique—SIT) or lethal mutation carriers. A second approach is population modification that leads to reduced risk of viral infection in the offspring. Mutations could be achieved through irradiation, chemical treatments, or advanced methods of genetic modification (e.g., gene-drive). In the SIT approach, it is expected that competition of sterile males with unmodified individuals could lead to the reduction in the local population by lower reproduction [174,180]. However, for the SIT technique to be effective, male mosquitoes have to be released multiple times. Therefore, improvements in mass mosquito production, precise sex separation, and release technologies are necessary to achieve cost efficiency [174]. The introduction of carriers of lethal mutations aim at producing progeny characterized by high mortality stemming, for example, from the production of females that lack the ability to fly [181].

Population modification aimed at reducing arbovirus transmission risk could be based on improving the natural anti-viral immunity of insects through RNA interference (RNAi). This approach was used in the construction of a transgenic *Aedes aegypti* strain characterized by high resistance to DENV-2 [143,182,183]. However, large-scale application of RNAi might be associated with some challenges, such as selection of an arbovirus quasispecies population that could lead to resistance [183,184]. Additionally, the optimization of the required amount of interfering dsRNA and high cost of its large volume production must be considered [185].

In summary, although innovative genetic prevention methods are promising, they remain controversial. Therefore, extensive field testing, law regulations, and negative attitudes of local communities may limit their wider use [183]. Nevertheless, those methods could be competitive with conventional prevention approaches [186].

## 7. Vaccines

Although they are considered as one of the most effective methods of infectious disease prevention, currently, vaccines directed against only a few of the major arboviruses are commercially available [187].

The first vaccine against YFV infection developed based on live attenuated strain 17D became broadly available in the 1930s. Currently used vaccines are based on the substrains of the same viral strain and, since the 1980s, only minor improvements have been made in its formulation [188]. Although, through decades of use, it proved to be highly effective in inducing an immune response, the occurrence of rare cases of vaccine-associated neurotropic and viscerotropic diseases have been reported. Additionally, as the vaccine is manufactured on embryonated chicken eggs, it is not recommended for people with hypersensitivity to egg proteins [189]. The labor-intensive production method of live attenuated vaccines is also problematic, as around 400 million people live in regions endemic to yellow fever, a number which exceeds global production capacities. For this reason, several alternative vaccines have been under development recently. In the first approach, the 17D strain of YFV inactivated using either beta-propiolactone or hydrogen peroxide is used to induce immunity. The second approach is based on the use of pre-membrane and envelope protein (preM/E), either as recombinant proteins or genes carried by a modified viral vector [35]. Additionally, the DNA construct encoding envelope protein of YFV has been tested, proving to be another promising vaccine candidate [190].

Contrary to YFV, DENV is characterized by high variability, with four known serotypes. Infection with one serotype results in lifelong immunity to that serotype but only temporal immunity to others. Additionally, although primary infection is usually asymptomatic, sequential infection with different serotypes often leads to severe forms of secondary infection such as dengue hemorrhagic fever (DHF) or dengue shock syndrome (DSS). This makes the development of an effective and safe vaccine more challenging as it would need to induce strong neutralizing antibodies to all serotypes at once without being highly pathogenic [191]. As a result, currently, only one live attenuated chimeric yellow fever 17D—tetravalent dengue vaccine (CYD-TDV)—Dengvaxia^®^ (Sanofi, Paris, France) has been licensed for use in some countries of Asia and Latin America [192]. This vaccine is based on modified vaccine strain 17D of YFV, in which genes encoding preM/E protein have been replaced with corresponding genes of four DENV serotypes [193]. Although it was proved that immunity induced by this vaccine lasts up to 4 years, its efficiency was dependent on virus serotype, host age, and serological status of the individual before vaccination. For example, it was shown that it is ineffective against serotype 4 of DENV, has reduced protectivity in previously seronegative individuals, and could lead to an increased risk of hospitalization in children under the age of nine. To overcome this limitation, several alternative vaccine formulas have been proposed, including other live attenuated, inactivated, recombinant subunits, as well as DNA vaccines [194]. However, currently, only two of them have reached the III phase of clinical trials. Those are TDV (Tetravalent Dengue Vaccine)—DENVax^®^ (Takeda Vaccines, Singapore) and TetraVax-DV, developed by the U.S. National Institute of Health (NIH). The first of them was based on live attenuated DENV-2 combined with preM/E genes from DENV-1, DENV-3, and DENV-4. The second one is based on two wild-type strains of DENV with attenuating deletions and one strain, rDENV4Δ30, additionally carrying the preM/E gene of DENV2. Thus far, they have both been shown to be safe and able to create immunity against all four serotypes of DENV [194].

Although infections of humans with ZIKV were recorded as early as the 1950s, only in the XXI century was it recognized as an important human pathogen. In 2016, the WHO declared that the zika virus was a public health emergency of international concern. For this reason, only recently attempts to develop a zika vaccine have been undertaken. At the moment, there are a number of vaccines under development, but all of them are at an early phase. The ideal vaccine should be safe for vulnerable populations and should enable the transfer of protective immunity to the developing fetus and newborn child without the neurological side effects observed with zika virus infections [195].

Currently, the only commercially available vaccines for WNV are licensed strictly for veterinary use. For any potential vaccine against WNV to be considered useful in humans, it is crucial that it is safe and induces strong immunity in elderly people, as infection in this group is most often connected with severe symptoms [196]. The development of WNV vaccines is slow as outbreaks of WNV are sporadic, which, as a consequence, requires a large number of volunteers, increasing total cost of testing. Among the six vaccines that are currently tested for human use, only one (ChimeriVax-WN02^®^, Sanofi, Paris, France) has reached the second phase of clinical trials. Analogically to the previously described dengue vaccine, it is based on the attenuated chimeric yellow fever 17D vaccine strain expressing the preM/E-fragment of WNV. It was shown to be safe for use in every age group and effective in over 90% of tested subjects [197,198].

Similarly, as in the case of WNV, the only commercially available RVFV vaccines are designated for veterinary use and licensed only in countries where the virus is endemic [199]. Nevertheless, RVFV inactivated vaccine TSI-GSD-200 (USAMRIID, Fort Detrick, MD, USA) was used for 10 years to protect a limited group of workers of the US Army considered to have a risk of infection, confirming its safety and immunogenicity for humans [200]. Due to the high costs and lack of broadly approved vaccines, wide-scale vaccination of humans is not practiced. However, routine vaccination of farmed animals coupled with vaccination of human risk groups is regarded as a potentially effective strategy in limiting the scale of infections with RVFV, as most human cases in endemic regions were directly or indirectly connected with the presence of virus in the livestock population [199].

As tick-borne encephalitis (TBE) is the most common arthropod transmitted viral infection in Europe as well as in central and eastern Asia, effective disease prevention via vaccinations is of utmost importance for public health in endemic countries [201]. Since its first description in the 1930s, multiple vaccine formulations have been developed and introduced for use [202]. Currently, five inactivated vaccines are available commercially: two in Europe, two in Russia, and one in China. Although different subtypes of tick-borne encephalitis virus (TBEV) were used for their creation, it was proven that vaccines formulated with one subtype are protective to infection with strains representing other subtypes. Large-scale vaccination proved to be an effective strategy in TBE prevention in countries where a significant part of the population was vaccinated, such as Austria (over 85%), which saw a dramatic decrease in the number of cases [201].

For prevention against Japanese encephalitis virus (JEV), another important arbovirus, four different vaccine types have been developed: mouse brain-derived inactivated, cell culture-derived inactivated, cell culture-derived live-attenuated, and genetically engineered live-attenuated chimeric. Mouse-brain-derived inactivated vaccine, JE-VAX^®®^ (Biken, Osaka, Japan), was the first one to be made available, however, although highly immunogenic, it had several drawbacks that led to the recent discontinuation. Its limitations included the risk of serious neurological side effects, high production costs, and the need for additional boosters. Cell culture-derived inactivated vaccine is currently used in many countries as a substitute to JE-VAX^®^ as it is similarly immunogenic but safer to use. Cell culture live-attenuated vaccine, used in China, is also characterized by high protection efficacy (99% after two doses) and safety, although, as in the case of all live vaccines, theoretical risk of reversion should be considered. The last type of currently available vaccine is a live-attenuated vaccine based on a chimeric YF-JE virus. It was designed analogically to the vaccines for DENV and WNV, described above, by combining the 17D yellow fever vaccine strain with genes encoding preM/E proteins of JEV. The most important benefits of using this vaccine are its ability to induce strong, long-lasting immunity after a single dose and the limited ability to replicate in mosquitos, preventing transmission from vaccinated person to other hosts [203].

Successful prevention of BTV infections in animals is challenging, as cross-protection against numerous serotypes of this virus is poor [204]. The importance of prevention against BTV became evident with the incursion of the BTV into Mediterranean Europe in the early 2000s. As a result, to limit losses and minimize virus spread, Italy, France, Portugal, and Spain undertook vaccination programs of their livestock that involved either only sheep or all domestic ruminants. Those programs were based either on modified live or inactivated vaccines, monovalent or polyvalent. Modified live vaccines were found to be cheap and effective in inducing immunity. However, their use was connected with the risk of teratogenicity, reversion to a virulent form, possibility of spread via insect vectors, and risk of restoration with wild-type viruses [205]. It was especially evident with some of the vaccines based on BTV-16, the use of which led to the typical symptoms of BT in sheep. In contrast, inactivated vaccines are safe, but their production is expensive and a booster is required to retain immunity [206]. Various types of viral vector vaccines have been designed to overcome those drawbacks. They used various viral vectors including poxviruses, adenovirus, or herpes viruses for delivery of BTV antigens. The use of RVFV as a viral vector was also tested, confirming its potential to induce immunity against both viruses simultaneously [207]. Viral vector vaccines against BTV were found to be not only safe and immunogenic but to also induce cross-reactivity against multiple serotypes of BTV and enable differentiation between vaccinated and infected animals [208].

As the number of emerging arboviruses of potentially high risk to human and animal health increase, successful prevention against multiple pathogens using vaccines becomes more costly and complicated. In recent years, alternative prevention solutions have been proposed based on using vaccines to target not one particular pathogen but salivary proteins of arthropods. Tests on mice showed that the presence of mosquito saliva in the place of WNV injection results in increased viral load [209]. A similar correlation was observed for other *flaviviruses* [210]. Several mechanisms were described that may explain this phenomenon, including suppression of antimicrobial peptide secretion by keratinocytes, inhibition of expression of type 1 IFN, cleavage of extracellular matrix proteins, and disruption of the communication link between the lymph nodes and surrounding stromal environment. Therefore, targeting vector saliva could prevent the creation of the microenvironment necessary for pathogen transmission [210]. Recently, the first human vector saliva-based phase 1 vaccine trial was completed using a *Anopheles gambiae* saliva vaccine (AGS-v) composed of four salivary peptides. The results showed its safety and immunogenicity; however, further field studies are required to investigate the ability of this kind of vaccine to provide long-lasting, meaningful protection against vector-borne diseases [211].

## 8. Conclusions

Arthropod-borne diseases are important from both medical and veterinary points of view. Arbovirus ecology is complex and often involves vector–animal–human interactions. Therefore, to combat the arboviral threats, it seems appropriate to use the “One Health” perspective, defined by U.S. Centers for Disease Control and Prevention as a: “collaborative, multisectoral, and transdisciplinary approach working at the local, regional, national, and global levels with the goal of achieving optimal health outcomes recognizing the interconnection between people, animals, plants, and their shared environment” [212]. The environmental aspects of disease control have often been neglected, but eradication of arboviral diseases particularly requires collaboration between human and animal health services, epidemiologists, entomologists, and environmentalists [11,213]. Actions should be taken to effectively control vectors, introduce vaccines against arboviruses, and facilitate access to diagnostic tests and appropriate medical care. Concern should be given to those arboviruses which re-emerge in new geographical areas. Additionally, as a result of climate change and globalization, those considered as endemic could become future threats to previously unaffected regions. The introduction of novel tools such as high-throughput metagenomic sequencing to identify circulating pathogens causing unusual diseases in humans and animals could be beneficial for arbovirus control. This approach, combined with a random environmental sampling of vectors, could provide an effective early warning system [214]. For arthropod vector controls, the Integrated Vector Management (IVM) approach should be considered to minimize costs and maintain ecological sustainability without compromising the effectiveness of preventive strategies [144]. Special care should be taken to develop strategies to control non-mosquito arboviral vectors, such as tick carriers of TBEV, as they have been relatively neglected, while growing evidence proves their importance. Additionally, as arthropod vectors often have wide host ranges, and transmission cycles of many arboviruses include both human populations and farmed animals, IVM planning should cover human and animal habitats alike. However, for this model to be functional, interdisciplinary cooperation is required under the One Health concept through integrated programs adequately funded at governmental and international levels.

## Figures and Tables

**Figure 1 jcm-11-03026-f001:**
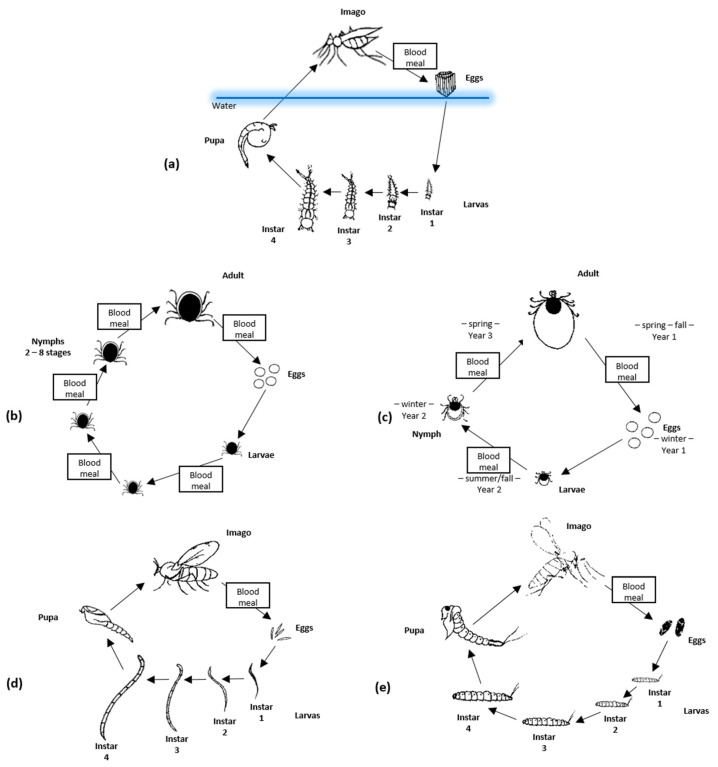
Life cycle of mosquitoes (**a**), soft ticks (**b**), hard ticks (**c**), biting midges (**d**), and sandflies (**e**).

**Figure 2 jcm-11-03026-f002:**
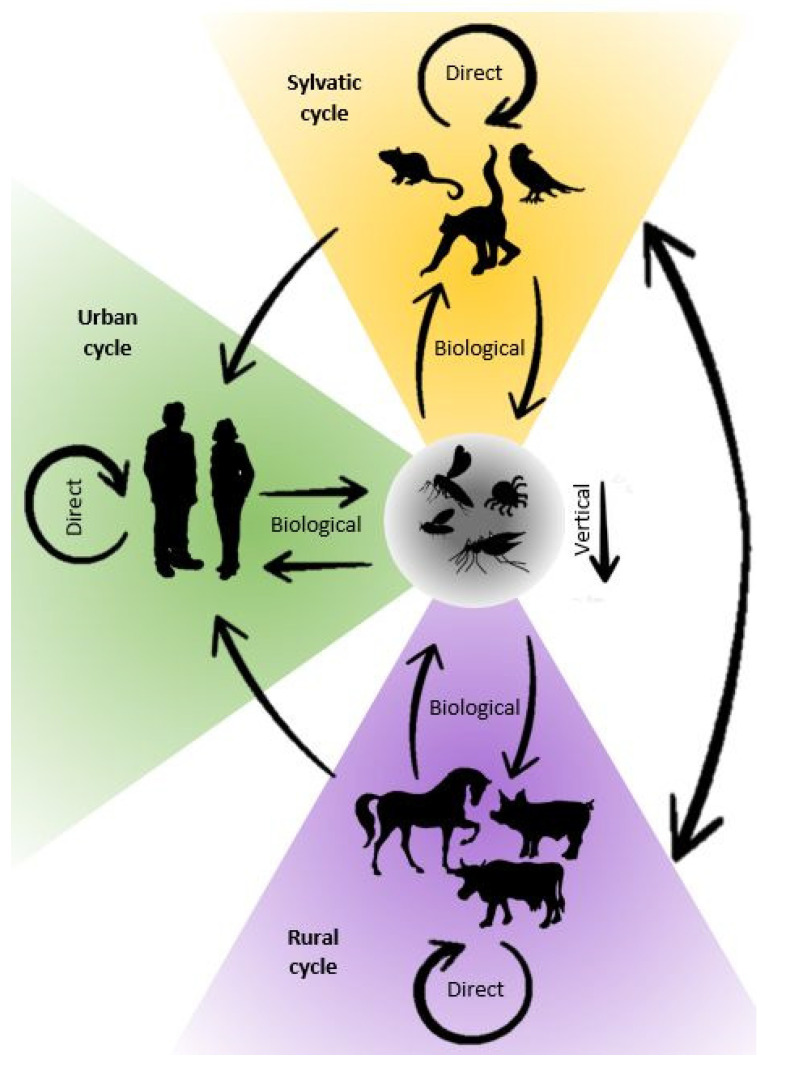
Environmental transmission cycles of arboviruses: sylvatic (yellow), urban (green), and rural (violet).

**Figure 3 jcm-11-03026-f003:**
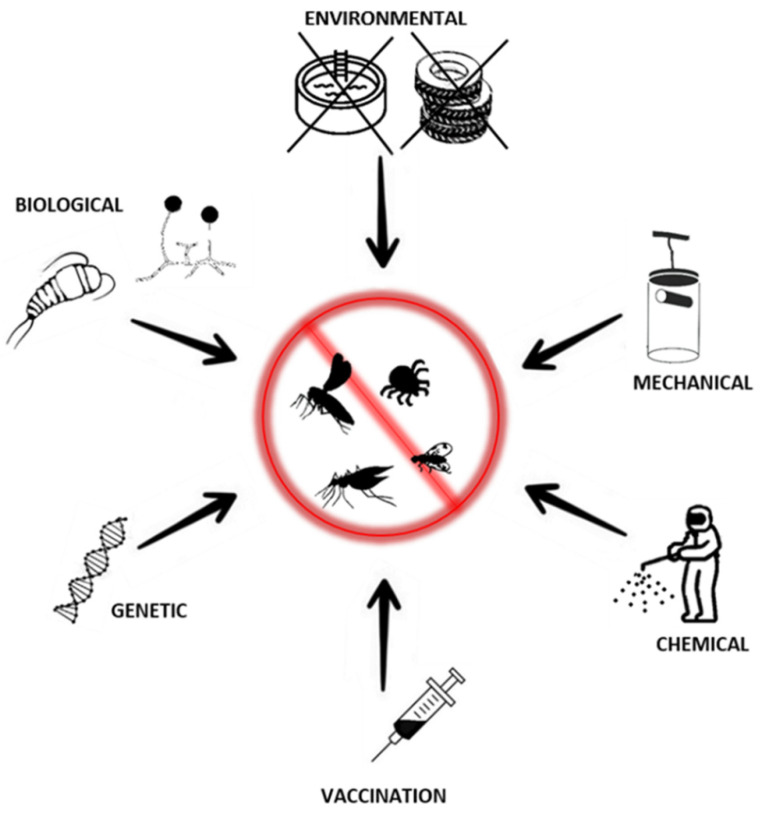
Prevention measures of arboviral disease control.

**Table 1 jcm-11-03026-t001:** Characteristic of selected arboviruses.

Virus	Vector	Main Hosts	Geographic Distribution	Scale Estimation	Symptoms	References
Family: *Flaviviridae*						
Dengue virus (DENV)	Mosquitoes	Monkeys, human	South and Central America, South Africa, the Arabian Peninsula, south Asia, Oceania	1,600,000 human cases (2021)	fever, frontal headache, myalgias and frequently arthralgias, nausea, vomiting and rash, vascular permeability, leakage, hypovolemia, shock	[12,33]
Yellow fever virus (YFV)	Mosquitoes	Primates, human	Africa central, central and northern part of South America	109,000 serious human cases (50,000 deaths in 2018)	fever, chills, generalized malaise, headache, red conjunctivae, photophobia, low back pain, myalgia, anorexia, nausea, vomiting, hepatomegaly, and epigastric and hepatic tenderness upon palpation, nausea, vomiting, epigastric pain, jaundice, oliguria, and hemorrhagic manifestations	[13,34,35]
Japanese encephalitis virus (JEV)	Mosquitoes	Pigs, birds, horses, human	Japan, East Asia, Indian Peninsula, Oceania	30–50,000 human cases/annually, 68,000 (2011)	cough, nausea, vomiting, diarrhea and photophobia, followed by a reduced level of consciousness, dull, flat, mask-like facies with wide, unblinking eyes, tremor, generalized hypertonia, cogwheel rigidity and other abnormalities of movement, motor neuron signs, cerebellar signs and cranial nerve palsies	[12,36,37]
Zika virus (ZIKV)	Mosquitoes	Apes, monkeys, human	North and South America, Pacific Asia, South and Central America	Central and North America—2725 human cases (2021) South America—18,318 human cases (2021)	fever, headache, rash, fetal abnormalities, headache, diffuse joint pain	[38,39]
West Nile Virus (WNV)	Mosquitoes	Birds, horses, human	Africa, Europe, Middle East, North America, East Asia	USA—2695 human cases, 191 fatalities (2021) EU/EEA—139 human cases (2021)	fever, headache, back pain, myalgias and anorexia, eye pain, pharyngitis, nausea, vomiting, diarrhea and abdominal pain can also occur	[12,40,41]
Family: *Togaviridae*						
Chikungunya virus (CHIKV)	Mosquitoes	Primates, human	Both Americas, Africa, Asia, central and eastern Europe, Oceania	225,000 human cases (2021)	fever, headache, prostration, conjunctival inflammation, myalgia, arthralgia, hemorrhagic signs, respiratory involvement, leukopenia, rash, lymphadenopathy, temp. may be diphasic	[33]
Ross River virus (RRV)	Mosquitoes	Marsupials, horses, human	Australia, Papua New Guinea, Islands in the Pacific Ocean	The highest number of infections in 1979–1980—more than 50,000 human cases. Australia—5000 human cases/year (2006–2015)	fever, chills, headache and aches and pains in the muscles and joints, swollen, rash	[42]
O’nyong nyong virus (ONNV)	Mosquitoes	Human, primates	Central, eastern and western Africa	No current data, 2 million human cases in East and West Africa (1959–1962)	fever, headache, prostration, myalgia, arthralgia, respiratory involvement, rash, lymphadenopathy	[43,44]
Family: *Phenuiviridae*						
Rift Valley fever virus (RVFV)	Mosquitoes, sandflies, midges	Cattles, buffalos, sheep, goats, camels, human	Arabian Peninsula, all of Africa with the exception of Côte d’Ivoire and northern countries (found in Egypt)	129 cases in humans and 109 in animals (2019) 3709 cases since 2000 Over 200,000 cases, including 598 fatalities in 1977–1978	fever, headache, prostration, conjunctival inflammation, stiffness, myalgia, arthralgia, CNS signs (including encephalitis, hemorrhagic signs, lymphadenopathy, vomiting, central scotoma-detached retina	[17,45]
Family: *Peribunyaviridae*						
California Encephalitis virus (CEV)	Mosquitoes	Human, rabbits, squirrels	California	About 68 human cases reported yearly	fever, headache, stiff neck, CNS signs (including encephalitis), CNS pleocytosis, and vomiting	[46,47]
Ngari virus (NRIV)	Mosquitoes, ticks	human, sheep, goats	Central Africa	No current data	fever, cold, sweating, headache, vomiting, nausea, diarrhea	[48,49]
Nyando virus (NDV)	Mosquitoes	Human	Central Africa	No current data	multiphasic fever, myalgia and vomiting	[50]
Pongola virus (PGAV)	Mosquitoes	Human	South Africa	No current data	fever, headache, joint pains	[50]
Schmallenberg virus (SBV)	Midges	Ruminants	Europe	No current data, 3444 infected herds (April 2012)	in cattle: fever, diarrhea, reduced milk yield, congenital malformation in newborn ruminants	[51]
Akabane virus (AKAV)	Mosquitoes, midges	Cattle, sheep, goats	Eastern hemisphere, including parts of Asia, Africa, the Middle East and Australia	Seroprevalence in Turkey: 44.74% cattle, 22.90%—sheep,14.52%—goats (2015–2017) China: 21.3%—cattle, 12.0%—sheep or goats (2006–2015)	in animals: congenital defects, abortions, stillborns, tremors, ataxia, lameness, paralysis, nystagmus, opisthotonos and hypersensitivity	[49,50,52,53]
Family: *Reoviridae*						
Bluetongue virus (BTV)	Midges, mosquitoes	Ruminants	Australia, North America, Africa, Middle East, Asia, Europe	EU—205 outbreaks (2021)	ruminants: fever, hyperaemia and congestion, leading to oedema mostly of the face, eyelids and ears, erosions of the mucous membranes, severe muscle degeneration, the lungs may show interalveolar hyperaemia	[20,23]
Epizootic hemorrhagic disease virus (EHDV)	Midges, mosquitoes	white-tailed deer, antelopes	North and South America, Africa, Asia, Australia, Indian Ocean islands	No current data	in animals: fever, anorexia, respiratory distress, oedema, conjunctivae, swelling of the tongue, oral/nasal erosions	[23,54]
African horse sickness virus (AHSV)	Midges	Horses, donkeys, mule, zebras	Sahara, Middle East, Turkey, southeast Asia	Outbreak in Pakistan and India (1959–1961)—more than 300,000 deaths. Outbreak in Thailand 2020 (610 infections, 568 deaths)	animals—fever, swelling of the supraorbital fossa, eyelids, facial tissues, neck, thorax, brisket and shoulders (subacute, oedematous or cardiac form), dyspnea, spasmodic coughing, dilated nostrils with frothy fluid oozing out (peracute, respiratory or pulmonary form)	[55,56]
Family: *Asfarviridae*						
African Swine Fever (ASFV)	Ticks	Pigs, bushpigs, warthogs	Central and eastern Europe, Italy, Belgium, Russia, China, south and central Africa with Madagascar	Global: 3762 outbreaks—domestic pigs (1,004,347 cases), 9229 outbreaks—wild boar (28,533 cases) (2021) EU: 1871 outbreaks—domestic pigs, 12,150 outbreaks—wild boar (2021)	animals—high fever, loss of appetite, haemorrhages in the skin and internal organ	[23,57]
Family: *Rhabdoviridae*						
Bovine ephemeral fever virus (BEFV)	Midges, mosquitoes	Bovine	Africa, Middle East, Australia, Asia	Seroprevalance in China: up to 81% (cattle from 26 of 28 provinces)	bovine: bi-phasic fever, salivation, lameness and muscle stiffness, general depression, muscle weakness, lameness and limp paralysis progressing to sternal recumbency	[32,58]
Family: *Orthomyxoviridae*						
Thogoto virus (THOV)	Ticks	Human, cattle, camel, antibodies in sheep and goat	Africa, Iran, southern Europe	Spain, seroprevalance: humans—5% among individuals with a history of tick bites) Sheep—20% (2020)	febrile illness accompanied by neurological symptoms in humans; afebrile leucopenia in cattle, and fever and abortion in sheep	[59,60]
Family: *Poxviridae*						
Lumpy skin disease virus (LSDV)	Mosquitoes, ticks	Cattle and wild ruminants	Africa, Middle East, Europe, Asia	Outbreaks: Iran—6, Iraq—8, Turkey—1294, Kazakhstan—1, Azerbaijan—16, Armenia—1, Russia—330 (2014–2016)	fever, inappetence, nasal discharge, salivation and lachrymation, enlarged lymph nodes, reduced milk production, loss of body weight, skin nodules on the neck, legs, tail, and back	[29]

## Data Availability

Not applicable.

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
