# Peer review of "Vector-Borne Viral Diseases as a Current Threat for Human and Animal Health—One Health Perspective"

_jcm, 2022, doi:10.3390/jcm11113026_

Round 1

Reviewer 1 Report

The stated objective of this manuscript is to review, from a One Health perspective, arboviral diseases, their transmission ecology and preventive measures related to a number of viruses transmitted by mosquitoes, ticks, biting midges, and sandflies.  Given the extensive information published regarding these topics over the past 5-10 years, this is an ambitious undertaking.  The authors provide a broad overview of selected arthropod-borne viruses of public health and animal health importance, summarizing their taxonomy, distribution, vector and host associations, symptoms, and provide generalized statements about their impact (i.e., human and animal cases).  The manuscript includes brief reviews of the life history patterns of the primary vectors, discussion of horizontal and vertical virus transmission and their contributions to virus maintenance and amplification.  Preventive measures are divided between those directed at reducing vector populations and vaccines directed at vector-transmitted pathogens, with the latter being one of the more detailed sections of the manuscript.    

Overall, the manuscript is concise and well written. What the manuscript lacks in detailed discussion in certain sections, it makes up for by providing numerous recent citations allowing readers to obtain more information where desired.  My main concern is that the title emphasizes addressing arboviral threats from a One Health perspective, yet One Health is only mentioned three other times: line 18 in the Abstract, line 49 in the Introduction, and line 698 in the Conclusion. While the manuscript does a good job reviewing the information, it does not address how to apply the One Health concept, as implied it will do in the Title and Abstract.  For example, the U.S. Centers for Disease Control and Prevention defines One Health as, “a collaborative, multisectoral, and transdisciplinary approach—working at the local, regional, national, and global levels—with the goal of achieving optimal health outcomes recognizing the interconnection between people, animals, plants, and their shared environment.”  Similarly, a recent article in  The Lancet (https://www.thelancet.com/journals/lanplh/article/PIIS2542-5196(18)30124-4/fulltext) states ,“One Health consists of the triad of human health, animal health, and the environment, but the latter is often neglected, as evident from its absence or cursory mention in most of the initiatives mentioned.”  I think the manuscript would benefit from a closing discussion about either how to promote prevention of arboviral disease through a One Health approach, or about how certain aspects of the multidisciplinary One Health approach must be strengthened to achieve relief from these disease transmission systems.  Otherwise, the cursory mentions of the phrase “One Health” are not meaningful in this manuscript.

Below are a few other comments the authors may wish to consider.

Introduction, Line 36.  Chikungunya is listed among the viruses causing encephalitis, however encephalitis is a rather rare manifestation of this virus, which is better characterized by a prolonged, debilitating arthritis/arthralgia.

Mosquitoes, Line 234.  The authors suggest the existence of a wingless mosquito species in Antarctica.  Actually, this insect (Belgica antarctica), is not a mosquito.  It is technically a midge and is a member of the family Chironomidae, not Culicidae, which includes the mosquitoes.

Mosqutoes, Line 246.  This sentence implies that Aedes mosquitoes, especially Aedes aegypti and Aedes albopictus are important vectors of the list of viruses that follows.  This is accurate for all of the viruses listed except West Nile Virus (WNV).  While WNV has been isolated from Aedes mosquitoes, mosquitoes in the genus Culex are the primary WNV vectors and there is no convincing information suggesting Aedes mosquitoes are important epidemic or epizootic vectors of WNV.

Chemical Prevention, lines 473-476.  This section states that the two main categories of insecticides are IGRs and chemical adulticides.  I suggest revising this section to identify the two main categories as larvicides (targeting the immature stages) and adulticides. IGRs are active ingredients that interfere with development of the immature stages.  Examples are methoprene and pyriproxifen. IGRs comprise only one type of larval control insecticide.  There are numerous other products widely used against immature stages of dipteran vectors.  These include products containing the bacteria Bacillus thuriengiensis israelensis and Lysinibacillus sphaericus which are consumed by the larval mosquitoes, and Spinosad which is an active ingredient derived from the bacteria Saccharopolyspora spinosa.

Biological prevention, lines 496-504.  Cyclopoid copepods are the only natural predator of mosquito larvae mentioned in this section.  Though they can be effective, they are rarely used in operational mosquito control programs.  I suggest the authors include mention of mosquito fish, such as Gambusia sp., which have been used much more widely in mosquito control programs.

Author Response

Revision 1

The stated objective of this manuscript is to review, from a One Health perspective, arboviral diseases, their transmission ecology and preventive measures related to a number of viruses transmitted by mosquitoes, ticks, biting midges, and sandflies.  Given the extensive information published regarding these topics over the past 5-10 years, this is an ambitious undertaking.  The authors provide a broad overview of selected arthropod-borne viruses of public health and animal health importance, summarizing their taxonomy, distribution, vector and host associations, symptoms, and provide generalized statements about their impact (i.e., human and animal cases).  The manuscript includes brief reviews of the life history patterns of the primary vectors, discussion of horizontal and vertical virus transmission and their contributions to virus maintenance and amplification.  Preventive measures are divided between those directed at reducing vector populations and vaccines directed at vector-transmitted pathogens, with the latter being one of the more detailed sections of the manuscript.    

Overall, the manuscript is concise and well written. What the manuscript lacks in detailed discussion in certain sections, it makes up for by providing numerous recent citations allowing readers to obtain more information where desired.  My main concern is that the title emphasizes addressing arboviral threats from a One Health perspective, yet One Health is only mentioned three other times: line 18 in the Abstract, line 49 in the Introduction, and line 698 in the Conclusion. While the manuscript does a good job reviewing the information, it does not address how to apply the One Health concept, as implied it will do in the Title and Abstract.  For example, the U.S. Centers for Disease Control and Prevention defines One Health as, “a collaborative, multisectoral, and transdisciplinary approach—working at the local, regional, national, and global levels—with the goal of achieving optimal health outcomes recognizing the interconnection between people, animals, plants, and their shared environment.”  Similarly, a recent article in  The Lancet (https://www.thelancet.com/journals/lanplh/article/PIIS2542-5196(18)30124-4/fulltext) states ,“One Health consists of the triad of human health, animal health, and the environment, but the latter is often neglected, as evident from its absence or cursory mention in most of the initiatives mentioned.”  I think the manuscript would benefit from a closing discussion about either how to promote prevention of arboviral disease through a One Health approach, or about how certain aspects of the multidisciplinary One Health approach must be strengthened to achieve relief from these disease transmission systems.  Otherwise, the cursory mentions of the phrase “One Health” are not meaningful in this manuscript.

Thank you very much for your insightful review, which notices the positive aspects of our work, and also for comments that allowed us to improve it, as we hope. We tried to broadly present the topic of arbovirus infections, seeing readers also in the group of human medicine doctors. The authors are mostly related to veterinary medicine and the presented issue is based on entomological data related to environmental sciences. Therefore, the concept of one health was clearly evident. We agree that this has not been sufficiently emphasized and summarized. Therefore, we propose the following changes in the "Conclusion" section:

starting at line 690

original

"Concern should be given both to those arboviruses which ranges is currently expending to new geographical areas, as well as to endemic arboviruses which often cause serious epidemics in affected regions. Additionally, as a result of climate change and globalization, those considered as endemic could become future threat to previously unaffected countries. Actions should be taken to effectively control vectors, introduce vaccines against arboviruses, and facilitate access to diagnostic tests and appropriate medical care. Arbovirus ecology is complex and often involves animal-human interactions. Therefore effective prevention and control of arbovirus infections demands introduction of One Health perspective."

corrected

"Arbovirus ecology is complex and often involves vector-animal-human interactions. Therefore, to combat the arboviral threats, it seems appropriate to use the "One Health" perspective defined by U.S. Centers for Disease Control and Prevention as: "collaborative, multisectoral, and transdisciplinary approach working at the local, regional, national, and global levels with the goal of achieving optimal health outcomes recognizing the interconnection between people, animals, plants, and their shared environment”[212]. The environmental aspects of disease control were often neglected but eradication of arboviral diseases particularly requires collaboration between human and animal health services, epidemiologists, entomologists and environmentalists [11,213]. Actions should be taken to effectively control vectors, introduce vaccines against arboviruses, and facilitate access to diagnostic tests and appropriate medical care. Concern should be given to those arboviruses which re-emerge in new geographical areas. Additionally, as a result of climate change and globalization, those considered as endemic could become future threat to previously unaffected regions. The introduction of novel tools such as high-throughput metagenomic sequencing to identify circulating pathogens causing unusual diseases in humans and animals could be beneficial for the arbovirus control. This approach combined with a random environmental sampling of vectors, could provide an effective early warning system [214]. However, for this model to be functional interdisciplinary cooperation is required under the One Health concept through integrated programs adequately funded at governmental and international levels."

We agree with all other comments of the Reviewer. We have made changes that we hope will meet the expectations of the Reviewer and raise the level of our manuscript.

Introduction, Line 36.  Chikungunya is listed among the viruses causing encephalitis, however encephalitis is a rather rare manifestation of this virus, which is better characterized by a prolonged, debilitating arthritis/arthralgia.

We agree with your comments, chikungunya was removed from hemorrhagic fevers, but included in arthritis.

Line 34

original

"Although arboviral infections may be asymptomatic or cause mild, transient influenza-like symptoms, they can be also associated with more severe consequences such as hemorrhagic fevers (e.g., dengue, yellow fever, chikungunya), encephalitis (e.g., Japanese encephalitis), or arthritis (e.g., Ross River fever, O'nyong-nyong fever)"

corrected

"Although arboviral infections may be asymptomatic or cause mild, transient influenza-like symptoms, they can be also associated with more severe consequences such as hemorrhagic fevers (e.g., dengue, yellow fever), encephalitis (e.g., Japanese encephalitis), or arthritis (e.g., Ross River fever, O'nyong-nyong fever, chikungunya)"

Mosquitoes, Line 234.  The authors suggest the existence of a wingless mosquito species in Antarctica.  Actually, this insect (Belgica antarctica), is not a mosquito.  It is technically a midge and is a member of the family Chironomidae, not Culicidae, which includes the mosquitoes.

We agree with your comment. We have removed fragment implying existence of wingless mosquitoes species on Antarctica.

original

"Different species of mosquitoes are identified from the tropics to Antarctica (wingless species) and the Arctic."

corrected

"Different species of mosquitoes are present on every continent except Antarctica."

Mosqutoes, Line 246.  This sentence implies that Aedes mosquitoes, especially Aedes aegypti and Aedes albopictus are important vectors of the list of viruses that follows.  This is accurate for all of the viruses listed except West Nile Virus (WNV).  While WNV has been isolated from Aedes mosquitoes, mosquitoes in the genus Culex are the primary WNV vectors and there is no convincing information suggesting Aedes mosquitoes are important epidemic or epizootic vectors of WNV.

As suggested, we removed the mention of WNV. Line 246:

original

"Aedes mosquitoes, especially Aedes aegypti and Aedes albopictus (Asian tiger mosquito) transmit dangerous viruses from families such as Flaviviridae, Togaviridae, Phenuiviridae including CHIKV, DENV, YFV, WNV, and RVFV"

corrected

"Aedes mosquitoes, especially Aedes aegypti and Aedes albopictus (Asian tiger mosquito) transmit dangerous viruses from families such as Flaviviridae, Togaviridae, Phenuiviridae including CHIKV, DENV, YFV, and RVFV"

Chemical Prevention, lines 473-476.  This section states that the two main categories of insecticides are IGRs and chemical adulticides.  I suggest revising this section to identify the two main categories as larvicides (targeting the immature stages) and adulticides. IGRs are active ingredients that interfere with development of the immature stages.  Examples are methoprene and pyriproxifen. IGRs comprise only one type of larval control insecticide. 

As suggested, we have introduced a division into larvicides and adulticides within the chemical prevention methods. Lines 473-474:

original

"Currently, two main categories of insecticides are used in vector control: insect growth regulators (IGRs) and chemical adulticides."

corrected 

"Currently, two main categories of chemical insecticides are used in vector control: larvicides (targeting the immature stages) including insect growth regulators (IGRs, e.g. pyriproxyfen, methoprene, diflubenzuron) and adulticides."

There are numerous other products widely used against immature stages of dipteran vectors.  These include products containing the bacteria Bacillus thuriengiensis israelensis and Lysinibacillus sphaericus which are consumed by the larval mosquitoes, and Spinosad which is an active ingredient derived from the bacteria Saccharopolyspora spinosa.

Accordingly, we included information on the topic of products containing the bacteria Bacillus thuringiensis israelensis and Lysinibacillus sphaericus and Spinosad. However, we have included this in the section on biological preventive measures.

Line 509 added:

"Another category of biological prevention methods are based on bacterial toxins that could act as natural larvicides (e.g. products of Bacillus thuringiensis svar. israelensis (Bti) and Lysinibacillus sphaericus) as well as fermentation product of Saccharopolyspora spinosa (Spinosan). They can be used either separately or in combination, which decreases the risk of resistance of insects and enables for wider target range"

Biological prevention, lines 496-504.  Cyclopoid copepods are the only natural predator of mosquito larvae mentioned in this section.  Though they can be effective, they are rarely used in operational mosquito control programs.  I suggest the authors include mention of mosquito fish, such as Gambusia sp., which have been used much more widely in mosquito control programs.

Accordingly, we included the information regarding larvivorous fishes used as biological prevention methods.

Line 499 added:

"The oldest example of biological control of mosquitoes, dates back to early 1900s when larvivorous fishes such as Gambusia affinis or Poecilia reticulata were artificially introduced as a part of control programs"

Once again, we would like to thank the Reviewer for his comments and effort put into evaluating our manuscript.

Reviewer 2 Report

When I was reading the entitled paper “Vector-borne viral diseases as a current threat for a human and animal health - One Health perspective” by Socha et al., I was really excited. The first four sections were “different”. Table 1 is fantastic. However, problems started at section 5, form this point the paper turns into “another nice vector control review with the same information” focusing in mosquitoes (specifically in Aedes aegypti or albopictus and dengue) control strategies.

Please, do not get me wrong; it is a well written paper, gathering nice and relevant information and I am aware that information about control of not-mosquito vectors still limited,  but authors promised us to “…focus on arboviruses infecting humans and animals, and their major vectors: mosquitoes, ticks, biting midges, and sandflies. The current knowledge on arbovirus transmission, ecology and ways of prevention are discussed” with a One Health Perspective approach, but authors did not accomplish the mission.  

I never found a discussion (control strategies are working? yes, no. All methods are cost-effective? For no-mosquito vectors, the same strategies could be used? Etc.).

Arbovirus ecology (only mentioned at line 696 but not discussed), how infecting different hosts affects, virulence or pathogenicity.  

The paper does not call physicians or veterinarians to unify their efforts in the management of vector-borne diseases, as contemplated under the One Health Perspective.

If authors want to publish a provocative manuscript I consider the paper must be improved.

Specific points:

Line 394-395. What authors meant with “genetically distinct cycles”, please explain and/or give examples.

Line 408. “…humans becoming a new reservoir”. Were you thinking about an organism that harbors the infectious agent without injury to itself and serves as a source from which other individuals can be infected? for example VIH. Or you meant an infective organism with a self-limiting arbovirus infection, such as dengue?

Maybe you meant a “new susceptibly host”?

Line  413. “…imported zoonoses could potentially "spillback" into local wildlife” Are any published information? can you give us an example?. I can´t deny is possible, but I am not aware of any paper showing a possible “spillback” zoonosis.

Rural cycles are as important as urban cycles? Please, consider public health and veterinary point of view.

Section 6.1 (starting at line 439) is well balanced. Maybe authors could discuss lines 448-450, why is difficult? You need a great number of interventions?

Related, environmental protection usually is performed in houses and it should be done by the residents, but residents expect health workers to do it. Please, discuss the restraints of environmental control strategies.

Section 6.2. Mechanical prevention (starting at line 460). Any mechanical device for not-mosquito vectors???

Section 6.3. Chemical prevention (starting at line 468). Depending in the application method (e.g. residual vs. ULV) could be argued that the strategy could be sometimes preventive, but also for outbreak control. Please discuss.

Related, all application methods used for mosquitoes are affective for other vectors? Or even among mosquito species? (Aedes vs. Anopheles).

Section 6.4. Biological prevention (starting at line 485). Entomopathogenic fungi could be used against all arbovirus-borne vectors? If it so good then why is not as used as products mentioned at section 6.3?

Wolbachia…cost-effective? compared with chemical prevention. Please discuss. When using Wolbachia, can we still use the other control methods? Wolbachia can limit dengue replication, can we expect viral evolution?

Section 6.4. Genetic control (starteing at line 528). Same as Wolbachia, is cost effective? Advantages and disadvantage of this strategy. Viral evolution against interference RNA, is possible?

Section 7. Vaccines. Nice!!!! Maybe lines 546-592 could be reduced.

Finally, the One Health Perspective is affordable or it is just a nice theoretical proposal? (Think in all the arboviral diseases mentioned in your ms).

Author Response

Reviewer 2

When I was reading the entitled paper “Vector-borne viral diseases as a current threat for a human and animal health - One Health perspective” by Socha et al., I was really excited. The first four sections were “different”. Table 1 is fantastic. However, problems started at section 5, form this point the paper turns into “another nice vector control review with the same information” focusing in mosquitoes (specifically in Aedes aegypti or albopictus and dengue) control strategies. Please, do not get me wrong; it is a well written paper, gathering nice and relevant information and I am aware that information about control of not-mosquito vectors still limited,  but authors promised us to “…focus on arboviruses infecting humans and animals, and their major vectors: mosquitoes, ticks, biting midges, and sandflies. The current knowledge on arbovirus transmission, ecology and ways of prevention are discussed” with a One Health Perspective approach, but authors did not accomplish the mission.  I never found a discussion (control strategies are working? yes, no.

All methods are cost-effective? For no-mosquito vectors, the same strategies could be used? Etc.).

            Thank you very much for your insightful review and appreciating the first part of the manuscript. We have made every effort to correct the second part of the manuscript. Due to the specificity of arboviruses and the threat they pose to both humans and animals, our goal was to write a review that would cover broad aspects of transmission, ecology and prevention. As the subject of arboviral infections is very wide we were forced to make some selections which areas we present in more detail and which we less extensively. Simultaneously we provided numerous recent citations allowing readers to obtain more information where desired.

            In our manuscript  we characterized the main arbovirus vectors. Since Aedes aegypti or Aedes albopictus are invasive species of mosquitoes and key vectors for viruses such as Chikungunya virus, dengue virus, yellow fever virus and Zika virus, we have devoted a lot of attention to them. In response to your comments we tried to expand the parts concerning other arthropod vectors, and to address some of the issues concerning different prevention strategies. We hope that the changes we have introduced will improve the quality of work and will make it more attractive to readers.

Arbovirus ecology (only mentioned at line 696 but not discussed), how infecting different hosts affects, virulence or pathogenicity. 

Our goal was to prepare a manuscript that will be useful in both human and veterinary medicine. We admit that no separate section was created to address the matter of arbovirus ecology. We have decided to focus more on the environmental transmission cycles involving arboviruses (described in section 5). However some of the issues concerning interaction between arboviruses, hosts and vectors were referred in section 3, lines 134-145. We are aware that manuscript would benefit from additional information on aspects of virulence and pathogenicity. We hope that the wide scope of this review excuse our decision to restrict details included in each of the sections.

The paper does not call physicians or veterinarians to unify their efforts in the management of vector-borne diseases, as contemplated under the One Health Perspective.

If authors want to publish a provocative manuscript I consider the paper must be improved.

We tried to broadly present the topic of arbovirus infections, seeing readers also in the group of human medicine doctors. The authors are mostly related to veterinary medicine and the presented issue is based on entomological data related to environmental sciences. Therefore, the concept of one health was clearly evident. We agree that this has not been sufficiently emphasized and summarized. Therefore, we propose the following changes in the "Conclusion" section:

starting at line 690

original

"Concern should be given both to those arboviruses which ranges is currently expending to new geographical areas, as well as to endemic arboviruses which often cause serious epidemics in affected regions. Additionally, as a result of climate change and globalization, those considered as endemic could become future threat to previously unaffected countries. Actions should be taken to effectively control vectors, introduce vaccines against arboviruses, and facilitate access to diagnostic tests and appropriate medical care. Arbovirus ecology is complex and often involves animal-human interactions. Therefore effective prevention and control of arbovirus infections demands introduction of One Health perspective."

corrected

"Arbovirus ecology is complex and often involves vector-animal-human interactions. Therefore, to combat the arboviral threats, it seems appropriate to use the "One Health" perspective defined by U.S. Centers for Disease Control and Prevention as: "collaborative, multisectoral, and transdisciplinary approach working at the local, regional, national, and global levels with the goal of achieving optimal health outcomes recognizing the interconnection between people, animals, plants, and their shared environment”[212]. The environmental aspects of disease control were often neglected but eradication of arboviral diseases particularly requires collaboration between human and animal health services, epidemiologists, entomologists and environmentalists [11,213]. Actions should be taken to effectively control vectors, introduce vaccines against arboviruses, and facilitate access to diagnostic tests and appropriate medical care. Concern should be given to those arboviruses which re-emerge in new geographical areas. Additionally, as a result of climate change and globalization, those considered as endemic could become future threat to previously unaffected regions. The introduction of novel tools such as high-throughput metagenomic sequencing to identify circulating pathogens causing unusual diseases in humans and animals could be beneficial for the arbovirus control. This approach combined with a random environmental sampling of vectors, could provide an effective early warning system [214]. However, for this model to be functional interdisciplinary cooperation is required under the One Health concept through integrated programs adequately funded at governmental and international levels."

Specific points:

Line 394-395. What authors meant with “genetically distinct cycles”, please explain and/or give examples.

We agree, that the phrase "genetically distinct cycles" was used in an insufficiently clear manner. What we meant, was that the mutations in viral genomes may lead to the more effective replication in a new host and in the transition between ecological cycles (e.g. VEEV transition from sylvatic to rural cycle). However, to avoid possible misunderstandings we removed this term from the text.

original

"As described by Weaver et al. 2021 [135], arbovirus transmission involves ecologically and genetically distinct cycles: the sylvatic (enzootic), the urban (human-amplified), or the rural (epizootic) (Fig. 2)"

corrected

"As described by Weaver et al. 2021, arbovirus transmission involves ecologically distinct cycles: the sylvatic (enzootic), the urban (human-amplified), or the rural (epizootic) (Fig. 2)"

Line 408. “…humans becoming a new reservoir”. Were you thinking about an organism that harbors the infectious agent without injury to itself and serves as a source from which other individuals can be infected? for example VIH. Or you meant an infective organism with a self-limiting arbovirus infection, such as dengue?

Maybe you meant a “new susceptibly host”?

We agree with your comment. The use of "new host" instead of "a new reservoir” is definitely more appropriate. Following change was introduced:

original

"However, the ability of anthropophilic mosquitoes such as Aedes aegypti to transmit these viruses has resulted in humans becoming a new reservoir and initiation of the urban cycle"

corrected

"However, the ability of anthropophilic mosquitoes such as Aedes aegypti to transmit these viruses has resulted in humans becoming a new host and initiation of the urban cycle"

Line  413. “…imported zoonoses could potentially "spillback" into local wildlife” Are any published information? can you give us an example? I can´t deny is possible, but I am not aware of any paper showing a possible “spillback” zoonosis.

This issue was described previously. For clarification we have added appropriate references:

  1. Chaves, A., Piche-Ovares, M., Ibarra-Cerdeña, C. N., Corrales-Aguilar, E., Suzán, G., Moreira-Soto, A., & Gutiérrez-Espeleta, G. A. (2021). Serosurvey of Nonhuman Primates in Costa Rica at the Human–Wildlife Interface Reveals High Exposure to Flaviviruses. Insects, 12(6), 554.
  2. Figueiredo, L. T. M. (2019). Human urban arboviruses can infect wild animals and jump to sylvatic maintenance cycles in South America. Frontiers in cellular and infection microbiology, 259.

Rural cycles are as important as urban cycles? Please, consider public health and veterinary point of view.

We have added two sentences that underline the importance of both perspectives.

added, starting at line 422

" The urban cycle is more important for human medicine as large urban centers are more prone to massive spread of infection. Rural cycles in areas with high livestock densities are the focus of veterinarians' attention. However, since there are viruses that can follow both cycles, they should be viewed together from the One Health perspective."

Section 6.1 (starting at line 439) is well balanced. Maybe authors could discuss lines 448-450, why is difficult? You need a great number of interventions?

To clarify we added following sentences starting at line 451:

"The specificity of the biology of ticks requires that each intervention be applied repeatedly and over a longer period of time. Additionally, unlike mosquitoes, tick breeding sites are more difficult to locate and remove"

Related, environmental protection usually is performed in houses and it should be done by the residents, but residents expect health workers to do it. Please, discuss the restraints of environmental control strategies.

We added sentences addressing this matter.

Starting at line 459:

"In general, for the environmental interventions to be effective, cooperation with local communities is crucial as for example mosquitoes source reduction requires removal of all potential breeding sites including those located on private proprieties. This is not possible without appropriate educational programs and/or hiring qualified professionals for surveillance."

Section 6.2. Mechanical prevention (starting at line 460). Any mechanical device for not-mosquito vectors???

We added some examples of challenges regarding the use of bednets and meshes for mechanical preventions of Culicoides and sandflies.

Starting at line 463

"Infusion with insecticides seems to be especially important for prevention against Culiccoides and sandflies as those vectors are small enough to pass through meshes and nets effective against mosquitoes, so purely mechanical barrier could be not effective"

Additionally we added an example of the use of mechanical prevention devices in limiting local tick population:

Starting at line 467:

"Similar approach was tested for limiting population of ticks Rhipicephalus sanguineus regarded mainly as a dog parasite but also able to transmit viruses (like THOV) to human. In that case, sticky traps baited with slowly released pheromones were found to reduce tick infestation in dog kennels"

Section 6.3. Chemical prevention (starting at line 468). Depending in the application method (e.g. residual vs. ULV) could be argued that the strategy could be sometimes preventive, but also for outbreak control. Please discuss.

We addressed this aspect of chemical prevention in sentences added starting at line 480"

"The choice of individual strategies depends on the desired goal as they can be used for prevention but also for outbreak control. For example, ground spraying with pyrethroids is effective in rapidly reducing number of female mosquitoes and thus effectively reducing the scope of the epidemic. However, in some countries its prophylactic use is limited due to the toxic effects"

Related, all application methods used for mosquitoes are affective for other vectors? Or even among mosquito species? (Aedes vs. Anopheles).

This problem was addressed in fragment added starting at line 481

"However, the choice of chemicals, its concentration and application techniques should be adapted to targeted vector species. For example, standard approach for control of mosquito population is distribution of adulticides as ultra-low volume sprays during the nighttime. While this approach is effective against Anopheles species (most active from dawn to dusk) it is less efficient for mosquito species with diurnal activity habits (e.g. Aedes aegypti or Aedes albopictus). This could potentially be overcome by the use of the adulticides that contain excitatory substances that force Aedes mosquitos to leave their nighttime resting places making them vulnerable to lethal aerosol [162]. From the other hand, this problem does not concern sandflies as they exhibit nocturnal activity and standard spraying procedure with pyrethroids could be adapted. However, as sandflies are terrestrial breeders and their larvae are difficult to locate, control measures are usually limited to those targeting adult individuals [156]. The use of chemical prevention against ticks, apart from aerosol spraying, includes baited traps designed to lure potential small mammalian hosts. Inside the traps animals are brushed with acaricids. This method was found to effectively reduce population of larval stages of ticks"

Section 6.4. Biological prevention (starting at line 485). Entomopathogenic fungi could be used against all arbovirus-borne vectors? If it so good then why is not as used as products mentioned at section 6.3?

More details concerning the use of entomopathogenic fungi was added starting at line 496

"Additionally they have to be carefully chosen for particular use e.g. Lagenidium giganteum, efficiently infects larvae of Aedes aegypti and Culex pipiens leading to high mortality but have no effect on Anopheles gambiae [166]. The efficiency of treatment also varies between groups of vectors. For example, while entomatopathogenic fungi were found to cause high mortality against each development stage of ticks, they were found to be effective predominately against larval stages of mosquitoes"

Wolbachia…cost-effective? compared with chemical prevention. Please discuss. When using Wolbachia, can we still use the other control methods? Wolbachia can limit dengue replication, can we expect viral evolution?

More details concerning the chances and challenges associated with the use of Wolbachia as a mean of biological prevention was added starting at line 518

"Efficiency of population suppression with Wolbachia is highly dependent on proper sex sorting which increases the cost of this approach"

and starting at line 525

"Currently, the most promising results have been obtained in the use of Wolbachia in limiting the spread of DENV by Aedes aegypti. In Indonesia, the sharp reduction of rate of Dengue fever cases was observed following release of Wolbachia infected mosquitoes [175]. It was found to be highly cost effective method which could give even better results when combined with vaccines [176]. Although viral evolution could lead to selection of resistant strains, multiple mutations in viral genome would be required to overcome broad mode of action of Wolbachia, without losing virus competence to both human and insect tissue"

Section 6.4. Genetic control (starteing at line 528). Same as Wolbachia, is cost effective? Advantages and disadvantage of this strategy. Viral evolution against interference RNA, is possible?

Whole section concerning genetic control was modified to address Reviewer concerns:

original:

" Genetic control methods are based on the mass release of genetically modified insect males of vector species into the environment. Modifications could either lead to sterility (so-called Sterile Insect Technique - SIT), mutations lethal for potential offspring, or limit the risk of viral infection. Mutations could be achieved through irradiation, chemical treatments, or advanced methods of genetic modification. In SIT approach it is expected that their competition with unmodified males could lead to the reduction of the local population by lower reproduction [171]. Other mutations aim at producing progeny characterized by high mortality stemming, for example, from the production of females that lack the ability to fly [172]. Another technique of genetic control that was proposed to limit arbovirus transmission risk is based on improving natural insect anti-viral immunity based on the RNA interference directed against viral RNA. This resulted, for example, in the construction of a transgenic Aedes aegypti strain characterized by high resistance to DENV-2 [144,173]."

corrected:

"Genetic control methods are based on the mass release of genetically modified insects into the environment. Modifications could lead to population suppression achieved through introduction of sterile males (so-called Sterile Insect Technique - SIT) or lethal mutation carriers. Second approach is population modification that leads to reduced risk of viral infection in the offspring. Mutations could be achieved through irradiation, chemical treatments, or advanced methods of genetic modification (e.g. gene-drive). In SIT approach it is expected that competition of sterile males with unmodified individuals could lead to the reduction of the local population by lower reproduction [174,180]. However for SIT technique to be effective male mosquitoes have to be released multiple times. Therefore improvements in mass mosquito production, precise sex separation and release technologies are necessary to achieve cost efficiency [174]. The introduction of carriers of lethal mutations aim at producing progeny characterized by high mortality stemming, for example, from the production of females that lack the ability to fly [181]. 

Population modification aimed at reducing arbovirus transmission risk, could be  based on improving natural anti-viral immunity of insects through the RNA interference (RNAi). This approach was used in the construction of a transgenic Aedes aegypti strain characterized by high resistance to DENV-2 [143,182,183]. However, large-scale application of RNAi might be associated with some challenges, like selection of arbovirus quasispecies population that could lead to resistance [183,184]. Additionally the optimization of the required amount of interfering dsRNA and high cost of its large volume production must be considered [184]. In summary, although innovative genetic prevention methods are promising, they remain controversial. Therefore extensive field testing, law regulations and negative attitudes of local communities may limit their wider use [183]. Nevertheless those methods could be competitive with conventional prevention approaches [186]"

Section 7. Vaccines. Nice!!!! Maybe lines 546-592 could be reduced.

We made some reductions in this section:

Lines 553-556

Line 589

Line 611

Lines 684-685

Finally, the One Health Perspective is affordable or it is just a nice theoretical proposal? (Think in all the arboviral diseases mentioned in your ms).

Issue of One Health Perspective was addressed in expanded "Conclusion" section.

Once again, we would like to thank the Reviewer for his comments and effort put into evaluating our manuscript.

Round 2

Reviewer 1 Report

The authors did a good job addressing all of the concerns I described in my review.  Nice work.

Author Response

Dear Reviewer, 

Thank you again for your comments and appreciation of our work. 

Wojciech Rozek 

Reviewer 2 Report

The paper has improved substantially.

At section 6, please, introduce the concept of Integrated Vector Management (IVM). State that control methods are not mutually exclusive, but they can act as complement of each other.

Also, at section 6, please, “warn” the audience that most of the methods have been developed for mosquito control, and explain the reasons.

At conclusions:

Call researches to investigate and develop control methods for non-mosquito vectors.

Finally, visit https://www.ivcc.com/vector-control/ivm/ and tried to match IVM with the One Health approach.

Author Response

Dear Reviewer,

Thank you very much for your comments. Accordingly we introduced changes.

 At section 6, please, introduce the concept of Integrated Vector Management (IVM). State that control methods are not mutually exclusive, but they can act as complement of each other.

Also, at section 6, please, “warn” the audience that most of the methods have been developed for mosquito control, and explain the reasons.

We added:

line 432

"Early prevention programs targeted mosquitoes and so far most of the actions have been directed against these vectors. This is due to their intense geographic expansion, the importance of the viruses they transmit, and thus the high risk to human life [95,142]."

line 439

"Those methods in most cases are not mutually exclusive and could act as complement of each other. The choice of particular methods or their combination should be based on Integrated Vector Management (IVM) - defined as a rational decision-making process to optimize the use of resources for vector control. This involves evaluation of cost effectiveness and ecological sustainability of prevention through collaboration with the health sector and local communities [144]."

At conclusions:

Call researches to investigate and develop control methods for non-mosquito vectors.

Finally, visit https://www.ivcc.com/vector-control/ivm/ and tried to match IVM with the One Health approach.

We added

line 775

"For arthropod vector controls, the Integrated Vector Management (IVM) approach should be considered to minimize costs and maintain ecological sustainability without compromising the effectiveness of preventive strategies [144]. Special care should be taken to develop strategies to control non-mosquitoes arboviral vectors, such as tick carriers of TBEV, as they have been relatively neglected while growing evidence proves their importance. Additionally, as arthropod vectors have often wide host ranged and transmission cycles of many arboviruses include both human population and farmed animals, IVM planning should cover human and animal habitats alike."

We hope that the introduced changes will meet the Reviewer's requirements. Thank You again for your effort.

With best regards,

Wojciech Rozek